# Rewiring an olfactory circuit by altering cell-surface combinatorial code

Cheng Lyu[1], Zhuoran Li[1,2], Chuanyun Xu[1,2], Jordan Kalai[1,2] & Liqun Luo[1✉]

Proper brain function requires the precise assembly of neural circuits during development. Despite the identification of many cell-surface proteins (CSPs) that help guide axons to their targets[1,2], it remains mostly unknown how multiple CSPs work together to assemble a functional circuit. Here we used synaptic partner matching in the *Drosophila* olfactory circuit[3,4] to address this question. By systematically altering the combination of differentially expressed CSPs in a single type of olfactory receptor neuron (ORN), which senses a male pheromone that inhibits male–male courtship, we switched its connection nearly completely from its endogenous postsynaptic projection neuron (PN) type to a new PN type that promotes courtship. From this switch, we deduced a combinatorial code including CSPs that mediate both attraction between synaptic partners and repulsion between non-partners[5,6]. The anatomical switch changed the odour response of the new PN partner and markedly increased male–male courtship. We generalized three manipulation strategies from this rewiring—increasing repulsion with the old partner, decreasing repulsion with the new partner and matching attraction with the new partner—to successfully rewire a second ORN type to multiple distinct PN types. This work shows that manipulating a small set of CSPs is sufficient to respecify synaptic connections, paving the way to investigations of how neural systems evolve through changes of circuit connectivity.

The precise wiring of neural circuits is the foundation of brain function. In his chemoaffinity hypothesis, Sperry speculated that "the cells and fibres of the brain and cord must carry some kind of individual identification tags, presumably cytochemical in nature, by which they are distinguished one from another almost, in many regions, to the level of the single neuron"[7]. Many CSPs have since been identified that guide axons to specific target regions[1,2]. CSPs that instruct synaptic partner selection within a specific target region have also begun to be identified[8]. However, disrupting individual CSPs, even with complete loss-of-function mutations, usually leads to partial phenotypes at specific wiring steps, particularly in synaptic partner selection[5,6,9], suggesting that there is considerable redundancy. Although redundancy could, in principle, increase the robustness of circuit wiring[3], it poses technical challenges to using a reductionist approach to achieve a complete understanding of how different CSPs work together to assemble a functional circuit—a central goal of developmental neurobiology.

An alternative approach to understanding circuit assembly is to re-engineer the combinatorial expression of CSPs in a single neuron type, with the aim of completely rewiring these neurons away from their endogenous synaptic partner and to a new partner. One of the challenges of rewiring a neural circuit is that the number of CSPs needed is, in general, thought to be large[8]. Here we report on such an approach in the *Drosophila* olfactory circuit.

In adult *Drosophila*, about 50 types of ORN form one-to-one synaptic connections with 50 types of PN at 50 discrete glomeruli, providing an excellent system for studying the mechanisms that underlie synaptic partner matching. Several previous studies have motivated our attempts to rewire the fly olfactory circuits. First, despite the three-dimensional organization of 50 glomeruli in adults, during development, each ORN axon only needs to search for synaptic partners along a one-dimensional trajectory on the surface of the antennal lobe[10]. This greatly reduces the number of synaptic partners among which individual ORN axons need to distinguish. Second, examining ORN axon development at single-neuron resolution revealed that each ORN axon extends multiple transient branches along its trajectory in early stages of development, and that branches that contact partner dendrites are selectively stabilized[4]. Third, in a companion manuscript[5], we describe the identification of three CSP pairs that signal repulsion during the partner matching process to prevent synaptic connections between non-cognate ORN and PN pairs. These repulsive CSPs, along with several attractive CSPs previously characterized[6] and reported here, are key components in the combinatorial codes for synaptic partner matching that we are about to describe.

## Genetic tools to visualize rewiring

We first sought to rewire ORNs that normally target their axons to the DA1 glomerulus (DA1-ORNs) to instead synapse with VA1v-PNs, the dendrites of which tile the VA1v glomerulus (Fig. 1a), by combinatorially manipulating the expression levels of different CSPs in DA1-ORNs. We chose these two glomeruli because the axons of both DA1-ORNs and VA1v-ORNs take similar trajectories during development[11], and because they process signals that have opposite effects on male courtship activity[12,13] (see details below). To simultaneously manipulate the

[1]Department of Biology and Howard Hughes Medical Institute, Stanford University, Stanford, CA, USA. [2]Biology Graduate Program, Stanford University, Stanford, CA, USA. ✉e-mail: lluo@stanford.edu

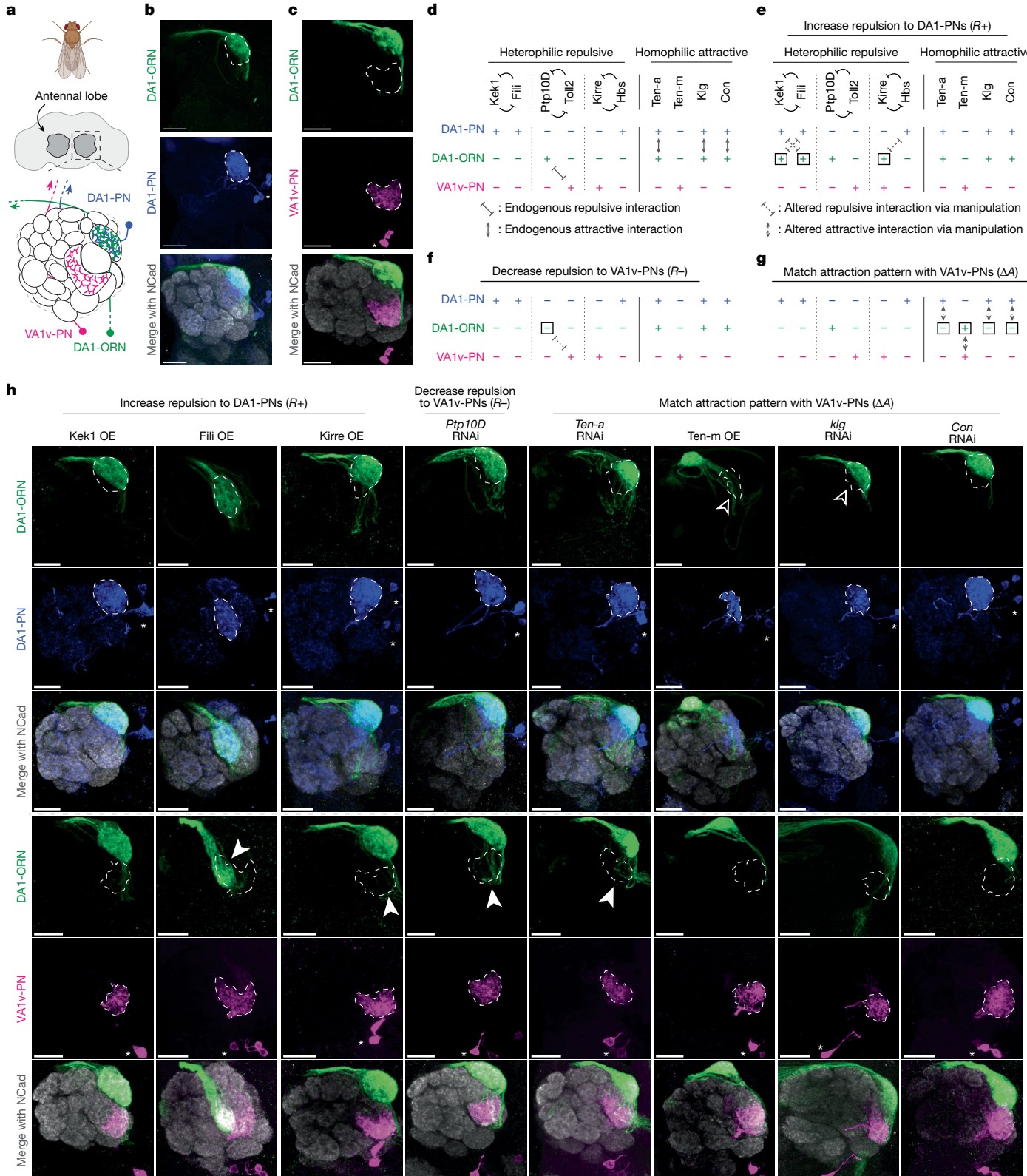

**Fig. 1** | See next page for caption.

expression levels of multiple CSPs only in DA1-ORNs during the wiring process, we generated a genetic driver that specifically labels DA1-ORNs across developmental stages using split-GAL4 (ref. 14) (referred to as the DA1-ORN driver; Extended Data Fig. 1). To examine the matching of DA1-ORN axons with the dendrites of either DA1-PNs or VA1v-PNs

in adults, we co-labelled DA1-ORNs (using the split-GAL4 above) with either DA1-PNs or VA1v-PNs in the same adult brain using the orthogonal QF/QUAS[15] and LexA/LexAop[16] systems, respectively (Fig. 1b,c). In wild-type flies, DA1-ORN axons overlapped with DA1-PN dendrites but not with VA1v-PN dendrites (Figs. 1b,c and 2a).

**Fig. 1 | Manipulating single CSPs in DA1-ORNs produces minor DA1-ORN→VA1v-PN rewiring. a**, Adult *Drosophila* brain and antennal-lobe schematics. DA1-ORN axons (green) match with DA1-PN dendrites (blue), but not with VA1v-PN dendrites (magenta). The same colour code is used in all other panels. **b**, Maximum *z*-projection of adult antennal lobes around DA1-ORN axons (green, labelled with a membrane-targeted GFP driven by a split-GAL4) and DA1-PN dendrites (blue, labelled with a membrane-targeted RFP driven by a QF2 driver). The DA1 glomerular border (dashed outline) was determined by N-cadherin (NCad) staining. Asterisks mark PN cell bodies. **c**, As in **b**, but with VA1v-PN dendrites (magenta) labelled instead of DA1-PN dendrites. The VA1v glomerular border is shown (dashed outline). **d**, Summary of expression levels of the ten CSPs in the rewiring experiments. '+' and '−' indicate relatively high and low expression levels, respectively, inferred mainly from the scRNA-seq dataset, and confirmed or corrected with the protein data when available (Extended Data Fig. 3). The endogenous expression patterns are shown at 24–30 h after puparium formation (APF), a developmental stage just before the onset of synaptic partner selection. **e**, Proposed genetic manipulations (in DA1-ORNs only) to increase the repulsion between DA1-ORN axons and DA1-PN dendrites during development. Square boxes in **e–g** indicate proposed genetic manipulations, with '+' for overexpression and '−' for RNAi knockdown. **f**, As in **e**, but for proposed genetic manipulations to decrease the repulsion between DA1-ORN axons and VA1v-PN dendrites. **g**, As in **e**, but for proposed genetic manipulations to match the attraction between DA1-ORN axons and VA1v-PN dendrites. **h**, Rewiring effects when CSPs are manipulated individually. Genetic manipulations are labelled at the top. Maximum *z*-projections of adult antennal lobes around DA1-ORN axons are shown. Top three rows: DA1-PNs are co-labelled with borders outlined (dashed lines). The open arrowheads indicate the decrease of overlap between DA1-ORN axons and DA1-PN dendrites. Bottom three rows (different brains from the top three rows): VA1v-PNs are co-labelled with borders outlined. Arrowheads indicate the mismatch of DA1-ORN axons with VA1v-PN dendrites. OE, overexpression. Overlapping ratios are quantified in Fig. 2a. Scale bars, 20 μm.

## Three manipulation strategies for rewiring

To achieve rewiring, we considered 10 CSPs that are likely to signal attractive or repulsive interactions during ORN–PN synaptic partner matching (Fig. 1d). Ten-a and Ten-m are type II transmembrane proteins that exhibit matching expression patterns across ORN and PN types and mediate homophilic adhesion[6]. Klingon (Klg) and connectin (Con) are also homophilic adhesion molecules involved in the development of the *Drosophila* visual and neuromuscular circuits, respectively[17,18]. On the basis of single-cell RNA sequencing (scRNA-seq) data[19,20], we found that Klg and Con also showed matching expression patterns across ORN and PN types (Extended Data Fig. 2). RNA interference (RNAi)-mediated knockdown[21,22] of *Con* and overexpression of Klg caused partial mismatching phenotypes consistent with their promoting homophilic attraction between ORNs and PNs (Extended Data Fig. 2). The remaining 6 CSPs form three groups—Kekkon 1 (Kek1) with Fish-lips (Fili), protein tyrosine phosphatase 10D (Ptp10D) with Toll2, and Kin of irre (Kirre) with Hibris (Hbs)—and signal repulsion between ORNs and PNs[5,9].

The expression levels of all CSPs were inferred mainly from scRNA-seq datasets during development[19,20]. Because the scRNA-seq data are prone to measurement noise and might not accurately reflect protein expression owing to post-transcriptional regulation, we corrected our RNA data using protein data and in vivo genetic manipulation results in CSPs for which additional data were available (Fig. 1d and Extended Data Fig. 3). As summarized in Fig. 1d, developing DA1-ORNs and DA1-PNs in the wild type contained attractive interactions from three CSPs (Ten-a, Klg and Con) but no repulsive interactions from the three repulsive pairs, in accordance with them forming synaptic partners in adults (Figs. 1d and 2a). By contrast, developing DA1-ORNs and VA1v-PNs contained no attractive interactions from the four attractive CSPs but repulsive interactions from one CSP pair (Ptp10D and Toll2) (Fig. 1b and Extended Data Fig. 3), consistent with them being non-synaptic partners in adults (Figs. 1c and 2a).

To facilitate rewiring, we used three genetic manipulation strategies during development, all restricted only to DA1-ORNs (Fig. 1e–g). (1) We increased repulsion between DA1-ORN axons and DA1-PN dendrites ('*R+*') to destabilize their interaction. Because the repulsive CSPs Kek1, Fili and Hbs are highly expressed in wild-type DA1-PNs, we overexpressed their interaction partners Fili, Kek1 and Kirre in DA1-ORNs (Fig. 1e). (2) We decreased repulsion between DA1-ORN axons and VA1v-PN dendrites ('*R−*') to stabilize their interaction. Because Ptp10D from DA1-ORNs mediates the repulsive interaction with Toll2 from VA1v-PNs in wild-type flies, we knocked down *Ptp10D* expression in DA1-ORNs (Fig. 1f). (3) We matched the expression pattern of attractive molecules between DA1-ORN axons and VA1v-PN dendrites ('*ΔA*') to stabilize their interactions and at the same time to destabilize the interactions between DA1-ORNs and DA1-PNs. Because the expression patterns of none of the four attractive CSPs between DA1-ORNs and VA1v-PNs match in wild-type flies, we genetically manipulated all four of them independently (Fig. 1g).

## Single-CSP changes cause minor rewiring

To start, we used the DA1-ORN driver (Extended Data Fig. 1) to over-express or knock down different CSPs in DA1-ORNs and to examine their individual effects on synaptic partner matching. All transgenes used in the repulsive interactions were validated in the companion study[5], and all transgenes used in the attractive interactions were either used in previous studies[6,17,18] or confirmed using multiple RNAi lines (Extended Data Fig. 2). Across the eight single-CSP manipulations (Fig. 1e–g), six showed observable but subtle DA1-ORN→VA1v-PN mismatching phenotypes (middle six columns in Fig. 1h, quantified in Fig. 2a and Extended Data Fig. 4), consistent with the results from previous manipulation experiments using these CSPs[4–6,9]. In the Ten-m-overexpression manipulation, most DA1-ORN axons no longer overlapped with DA1-PN dendrites, but none of the mistargeted DA1-ORN axons overlapped with VA1v-PN dendrites. This is consistent with the previous finding that Ten-m-overexpressing DA1-ORN axons are most likely to mismatch with DL3-PN dendrites[4]. This could be because the CSP profile of DL3-PNs matches with the profile of DA1-ORNs (after Ten-m overexpression) better than it does with that of DA1-PNs.

## Combinatorial changes enhance rewiring

Next, we simultaneously manipulated the expression of multiple CSPs in DA1-ORNs. We first aimed to find the CSP combination within each of the three manipulation strategies described above (*R+*, *R−* and *ΔA*; Fig. 1e–g) that can most strongly decrease the overlap between DA1-ORN axons and DA1-PN dendrites (loss of innervation, LoI) and increase the overlap between DA1-ORN axons and VA1v-PN dendrites (gain of innervation, GoI). Overexpression of both Kek1 and Fili ('*R+ ×2*' in Fig. 2a,b) led to a significant LoI and a significant GoI (Extended Data Fig. 4) compared to overexpressing either alone. This was the strongest phenotype that we observed among the different combinations of overexpressing repulsive CSPs. For example, overexpressing all three repulsive CSPs (Kek1, Fili and Kirre, '*R+ ×3*' in Fig. 2a,b) improved neither GoI nor LoI compared to '*R+ ×2*' (Extended Data Fig. 4). Therefore, we chose overexpressing Kek1 and Fili ('*R+ ×2*') as the best combination for the strategy of increasing repulsion between DA1-ORNs and DA1-PNs. Similarly, for the strategy of matching the expression pattern of attractive molecules between DA1-ORNs and VA1v-PNs, we found that knocking down *Ten-a* and *Con* simultaneously yielded the most significant LoI and GoI ('*ΔA ×2*' in Fig. 2a,b). We chose knocking down *Ptp10D* ('*R−*' in Fig. 2a,b) as the strategy of decreasing the repulsion of DA1-ORNs and VA1v-PNs, because it is the only relevant manipulation.

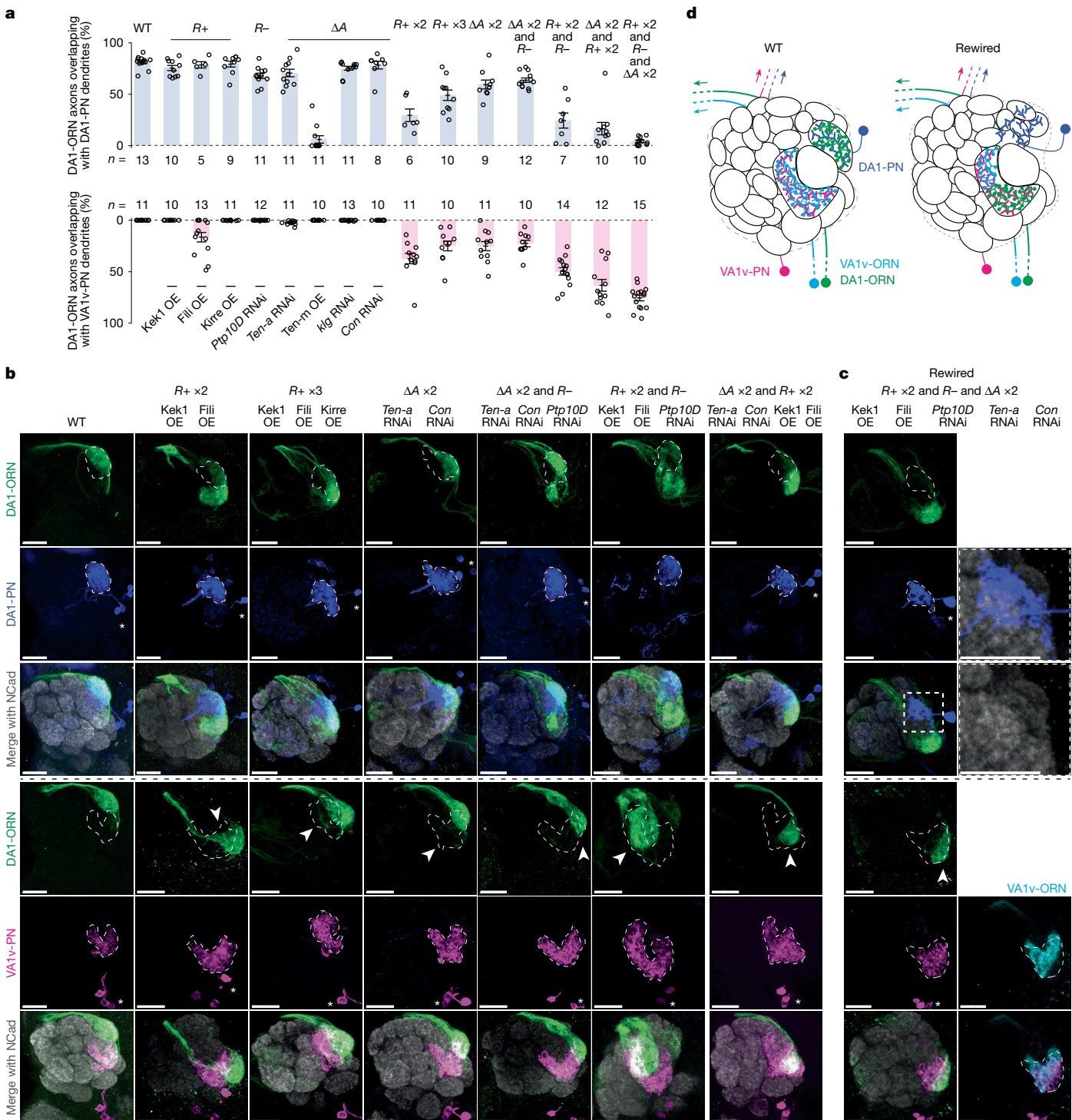

**Fig. 2 | Simultaneously altering the expression of five CSPs in DA1-ORNs causes a nearly complete rewiring of DA1-ORNs to VA1v-PNs. a**, Percentage of DA1-ORN axons overlapping with the dendrites of DA1-PNs (top) and VA1v-PNs (bottom). Circles indicate individual antennal lobes; bars indicate the population mean ± s.e.m. '*R*+ ×2': Kek1 OE + Fili OE. '*R* + ×3': Kek1 OE + Fili OE + Kirre OE. '*ΔA* ×2': *Ten-a* RNAi + *Con* RNAi. WT, wild type. **b**, Rewiring effects when CSPs are manipulated combinatorially. Genetic manipulations are labelled on the top. Maximum *z*-projections of adult antennal lobes around DA1-ORN axons (green) are shown. Top three rows: DA1-PNs (blue) are co-labelled with borders outlined (dashed lines). Bottom three rows: VA1v-PNs (magenta) are co-labelled with borders outlined (dashed lines). Arrowheads indicate the mismatch of

DA1-ORN axons with VA1v-PN dendrites. Asterisks mark PN cell bodies. Overlapping ratios are quantified in **a**. The leftmost column is a repeat of Fig. 1b,c for ease of comparison within this panel. **c**, Same as **b**, but with all three manipulation strategies combined. The two images at the top of the right column are magnifications of the dashed squares to the left. The two images at the bottom of the right column are from the same brain as in the left column, but with VA1v-ORNs co-labelled (cyan, *Or47b*-promotor-driven membrane marker). **d**, Summary of DA1-ORNs and DA1-PNs, as well as VA1v-ORNs and VA1v-PNs, in the wild-type (left) and DA1-ORN-rewired (right) antennal lobe. In the rewired lobe, DA1-ORN and VA1v-ORN axons split VA1v-PN dendrites; DA1-PN dendrites spread into multiple adjacent glomeruli. Scale bars, 20 μm.

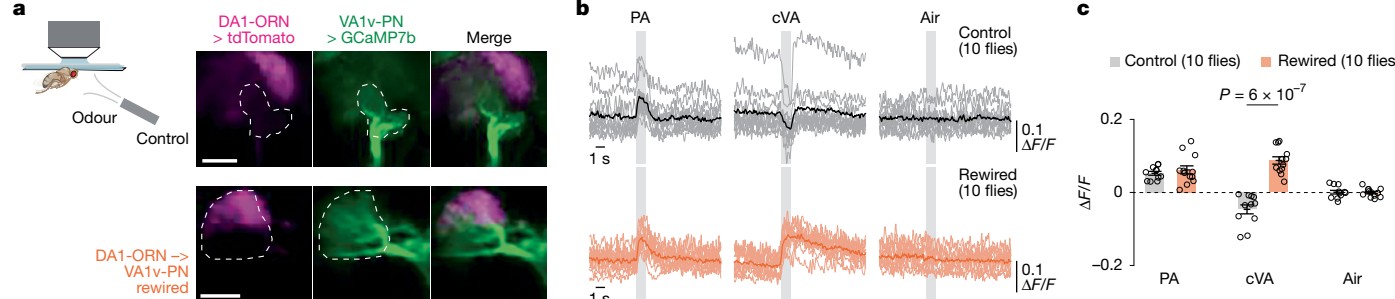

**Fig. 3 | VA1v-PNs in DA1-ORN-rewired flies respond to both VA1v- and DA1-specific odours. a**, Imaging neural activity in a plate-tethered fly with odorized air flow delivered to the fly antennae. Images of tdTomato signal in DA1-ORN axons and GCaMP7b signal in VA1v-PN dendrites are shown from a control fly (top) and a rewired fly (bottom). Images are averaged across the entire recording. The VA1v glomerulus is outlined according to the GCaMP7b signal. The imaging angle here is from dorsal to ventral, whereas in all other images it is from anterior to posterior. Scale bars, 20 μm. **b**, Averaged GCaMP7b activity in VA1v-PN dendrites in response to odorized air flows, measured by fluorescence intensity change over baseline ($\Delta F/F$). Grey vertical stripes indicate odorized air flows (1 s each). Light and dark traces indicate the means of individual flies and the population mean, respectively. In wild-type flies, the fly pheromone PA specifically activates VA1v-ORNs[12] and the fly pheromone cVA specifically activates DA1-ORNs[13,28,29]. **c**, Change of GCaMP7b activity in VA1v-PN dendrites in response to odorized air flows, calculated by subtracting the average GCaMP7b activity in the 0.5 s before odour delivery onset from that in the last 0.5 s of odorized airflow. Circles indicate the means of individual flies; bars indicate the population mean ± s.e.m. Unpaired two-sided $t$-test.

Next, we combined the best options from the three manipulation strategies. When we used two strategies simultaneously ('$\Delta A \times 2$ & $R-$', '$\Delta A \times 2$ and $R+ \times 2$', and '$R+ \times 2$ and $R-$' in Fig. 2a,b), in most cases, the LoI further decreased and the GoI further increased, compared with each strategy alone. For example, in '$\Delta A \times 2$ & $R+ \times 2$', the LoI was significantly more severe than was the LoI in either '$\Delta A \times 2$' or '$R+ \times 2$', and the GoI in the combined group was also significantly larger than was the GoI from each group (Fig. 2a,b and Extended Data Fig. 4).

When we combined all three manipulation strategies, nearly all DA1-ORN axons disconnected with DA1-PN dendrites and overlapped with VA1v-PN dendrites ('$R+ \times 2$ and $R-$ and $\Delta A \times 2$' in Fig. 2a,c,d). Dendrites of DA1-PNs seemed to spread into multiple adjacent glomeruli (inset in Fig. 2c), potentially forming synaptic connections with new ORN partners[10]. Furthermore, DA1-ORN axons only overlapped with part of VA1v-PN dendrites (bottom of Fig. 2c). We confirmed that the non-overlapping part of VA1v-PN dendrites matched with their natural partner VA1v-ORN axons (bottom of Fig. 2c), presumably because we did not genetically manipulate either VA1v-ORNs or VA1v-PNs. Notably, the axons of DA1-ORNs and VA1v-ORNs are segregated in the rewired flies (Fig. 2c), suggesting potential axon–axon repulsive interactions, as previously shown in a different context[23].

In this final rewiring experiment (referred to hereafter as DA1-ORN-rewired flies), the expression levels of five CSPs were changed in DA1-ORNs (Kek1, Fili, Ptp10D, Ten-a and Con; Fig. 2c). When any one of the five CSP changes was omitted, the rewiring was less complete (Extended Data Fig. 5). Although the DA1 glomerulus is sexually dimorphic in size[24,25], the DA1-ORN→VA1v-PN rewiring showed similar levels of change in male and female flies (Extended Data Fig. 6). Moreover, axons of several additional types of ORNs remained confined within their original glomeruli in rewired flies (Extended Data Fig. 6), supporting that the rewiring is specific to the DA1 and VA1v glomeruli.

### Rewiring alters the VA1v-PN odour response

To examine whether the anatomical DA1-ORN→VA1v-PN rewiring is accompanied by the formation of functional synaptic connections, we measured the neural response of VA1v-PN dendrites to VA1v- or DA1-specific odours in tethered flies (Fig. 3a). All ORN–PN connections are excitatory and use the same cholinergic neurotransmitter system[26]. We used the LexA/LexAop system to express GCaMP7b in VA1v-PNs, and measured intracellular $Ca^{2+}$ concentrations through two-photon excitation of GCaMP7b[27] as a proxy for neural activity. We simultaneously expressed and co-imaged tdTomato in DA1-ORNs

with GCaMP7b and confirmed the occurrence of DA1-ORN→VA1v-PN rewiring in these flies (Fig. 3a).

We next tested the odour responses of VA1v-PN dendrites. The pheromone 11-*cis*-vaccenyl acetate (cVA) specifically activates DA1-ORNs in the fly antennal lobe[13,28,29], and palmitoleic acid (PA) is a fly cuticular pheromone that specifically activates VA1v-ORNs in the fly antennal lobe[12]. In wild-type flies, we found that the activity of the dendrites of VA1v-PNs increased in response to PA and decreased in response to cVA (Fig. 3b,c). The inhibitory response of VA1v-PNs to cVA in wild-type flies is consistent with the previously described lateral inhibition from local interneurons (LNs) in the fly olfactory circuit[30,31]. In the rewired flies, however, both PA and cVA activated VA1v-PNs (Fig. 3b,c), supporting functional synaptic connections between DA1-ORN axons and VA1v-PN dendrites. We cannot rule out the possibility that altered connectivity of LNs, which exhibit diverse anatomical patterns[32,33], also contributes to the altered odour response. However, the inhibitory response of VA1v-PNs to odours that do not strongly activate VA1v- or DA1-ORNs remained similar between the rewired flies and the wild-type flies (Extended Data Fig. 7), suggesting that the connection between VA1v-PNs and LNs remained largely unchanged.

### Rewiring promotes male–male courtship

We next investigated whether DA1-ORN→VA1v-PN rewiring led to any behavioural changes in flies. In *Drosophila melanogaster*, cVA is only produced in males and acts through the Or67d odorant receptor, expressed in DA1-ORNs, to inhibit the courtship of males towards other males or recently mated females[13] (owing to cVA transferred from males to females during copulation[34]). The pheromone PA, on the other hand, promotes courtship in males through the Or47b odorant receptor expressed in VA1v-ORNs[12]. Therefore, in rewired flies, a pheromone that normally inhibits male–male courtship (cVA) now activates a pathway (VA1v) that promotes courtship. This suggests that rewired males might attempt to court other males.

To test this prediction, we introduced two virgin males—one wild type and one with DA1-ORN rewired—into the same behavioural chamber (Fig. 4a). We then recorded video for 25 min and analysed the unilateral wing extension events of both makes (Fig. 4b and Supplementary Videos 1 and 2). This is a typical male courtship behaviour during which males vibrate one of their wings to produce courtship song[35]. We found that the rewired males exhibited unilateral wing extensions towards their wild-type partner males significantly more frequently than the other way around (Fig. 4c,d). In a separate experiment, we introduced

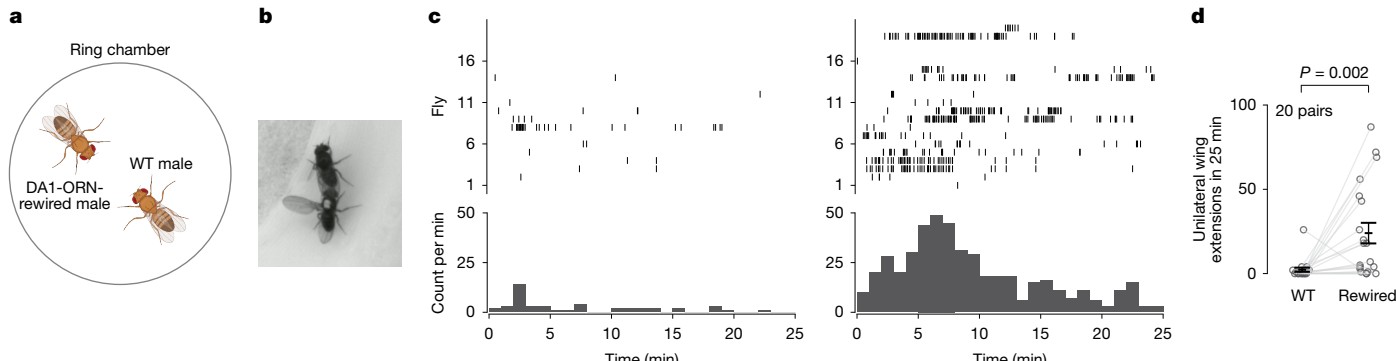

**Fig. 4 | DA1-ORN-rewired male flies show increased courtship activity towards other males. a**, Courtship assay. One wild-type male and one DA1-ORN-rewired male are introduced in the same behavioural chamber (diameter 2 cm) to monitor their courtship activity towards each other. **b**, Example frame of a unilateral wing extension from a DA1-ORN-rewired male (white dot on the thorax) towards a wild-type male. **c**, Rasters of unilateral wing extensions (top) and extension count per minute (bottom) exhibited by wild-type (left) and rewired (right) males. Fly numbers in the right and left panels denote the same fly pairs. **d**, Total number of unilateral wing extensions in 25-min recordings. Circles indicate individual flies; bars indicate the population mean ± s.e.m. Two-sided Wilcoxon signed-rank test.

one male—either wild type or rewired—with a virgin female into the behavioural chamber. We did not observe any detectable differences between wild-type and rewired males in courtship activity towards virgin females (Extended Data Fig. 8a–c). This is consistent with our working model, because a virgin female does not have cVA, and the connections between VA1v-ORNs and VA1v-PNs in rewired flies remained intact, as assayed anatomically (Fig. 2c) and physiologically (Fig. 3b,c). Experimental silencing or activation of DA1-ORNs in rewired males further revealed that both the loss of connection to DA1-PNs and the gain of connection to VA1v-PNs in rewired males contributed to the increased male–male courtship activity (Extended Data Fig. 8d–l). Finally, when five virgin rewired males were introduced into the same behavioural chamber, they exhibited vigorous chasing and courtship activities, sometimes forming a courtship chain in which a male attempted to court the male in front of him while being courted by another male behind him (Supplementary Video 3).

## Generalization to other glomeruli

We wondered whether the same set of CSPs and wiring strategies apply to the ORN–PN synaptic partner matching in other glomeruli. To test this, we aimed to rewire the axons of another ORN type, VA1d-ORNs, to the dendrites of PNs targeting three distinct neighbouring glomeruli: VA1v, DC3 and DL3 (Fig. 5).

We used a genetic driver that specifically labels VA1d-ORNs across developmental stages using split-GAL4 (ref. 4), and simultaneously labelled the dendrites of VA1d-PNs, VA1v-PNs, DC3-PNs or DL3-PNs in the same adult brain using the orthogonal LexA/LexAop[16] or QF/QUAS[15] systems (Fig. 5b). In wild-type flies, VA1d-ORN axons overlapped with the dendrites of VA1d-PNs almost exclusively, and showed minimal overlap with the dendrites of other PN types (Fig. 5b,c). The goal of rewiring is to switch the axons of VA1d-ORNs to match with the dendrites of each of the three other PN types in separate experiments.

On the basis of the same 10 CSPs described above, during the development of wild-type flies, VA1d-ORN axons and VA1d-PN dendrites form two attractive interactions (through Ten-m and Con) and no repulsive interactions (Fig. 5a and Extended Data Fig. 9). For the first manipulation strategy, which aims to increase the strength of repulsion between VA1d-ORNs and VA1d-PNs, we could overexpress the repulsive CSPs Kek1, Toll2, Kirre or Hbs in all three rewiring attempts (Fig. 5a, top). For the second manipulation strategy, which aims to decrease the strength of repulsion between VA1d-ORNs and other PN types, we sought to knock down *Ptp10D* in two of the three switch attempts and do nothing in the switch attempt to DL3-PNs, because DL3-PNs do not exhibit any repulsive interactions with VA1d-ORNs from these three

repulsive pairs (Fig. 5a, bottom left). For the third manipulation strategy, which aims to match the expression pattern of attractive molecules between VA1d-ORNs and other PN types, we could overexpress or knock down the expression of these four attractive CSPs accordingly (Fig. 5a, bottom right).

Using the different combinations of manipulations described above, we were able to rewire more than half of VA1d-ORN axons to match with the dendrites of either VA1v-PNs or DC3-PNs in two separate experiments, and to rewire almost all VA1d-ORN axons to match with DL3-PN dendrites in a third experiment (Fig. 5b,c). In all three rewiring experiments, the part of VA1d-ORN axons that did not match with the dendrites of target PNs remained matching with the dendrites of their natural partner VA1d-PNs (Fig. 5b,c). Note that in the rewiring to VA1v-PNs and DL3-PNs, we also included an additional manipulation: *Sema2b* knockdown. This is because VA1d-ORNs have a higher expression level of *Sema2b* than do VA1v-ORNs and DL3-ORNs[19]. Given this, we speculated that VA1v-ORN and DL3-ORN axons take a more dorsolateral trajectory than do VA1d-ORN axons when they sweep through the antennal-lobe surface. Because a single ORN axon searches mainly in the vicinity of their trajectory[4], we included *Sema2b* knockdown to shift the axons of VA1d-ORNs more dorsolaterally[10,11] so that their trajectories could be closer to the dendrites of VA1v-PNs and DL3-PNs. Consistently, when all the manipulations remained the same, but the *Sema2b* knockdown was left out, there was less matching between VA1d-ORN axons and VA1v- or DL3-PN dendrites (Extended Data Fig. 10).

To test whether the anatomical rewiring of VA1d-ORNs described above leads to the formation of functional synaptic connections, we examined in rewired flies whether the different PN types would gain responses to VA1d-ORN-specific odours, the pheromones methyl palmitate (MP)[28] or methyl myristate (MM)[12] (see Fig. 5 legend for more detail). Using the same set-up as in Fig. 3a, we measured the neural response of VA1v-, DC3- and DL3-PNs, separately, through two-photon excitation of GCaMP variants expressed using the LexA/LexAop or QF/QUAS system in these PNs (Fig. 5d–i). We also co-expressed tdTomato in VA1d-ORNs to confirm the anatomical switch in these flies. In all three rewiring experiments, the dendrites of target PNs had a stronger response to VA1d-ORN-specific odours, compared with wild-type flies (Fig. 5d–l). Note that in the case of DL3 (Fig. 5i,l), although rewiring eliminated the inhibitory response of DL3-PNs to MP, the magnitude of the positive response was much smaller. We speculate that this could result from the substantial lateral inhibition that DL3-PNs might receive from other MP-responding ORN types. Altogether, these results show that the three genetic strategies for altering cell-surface combinatorial code are generalizable for selecting synaptic partners in the fly olfactory circuit.

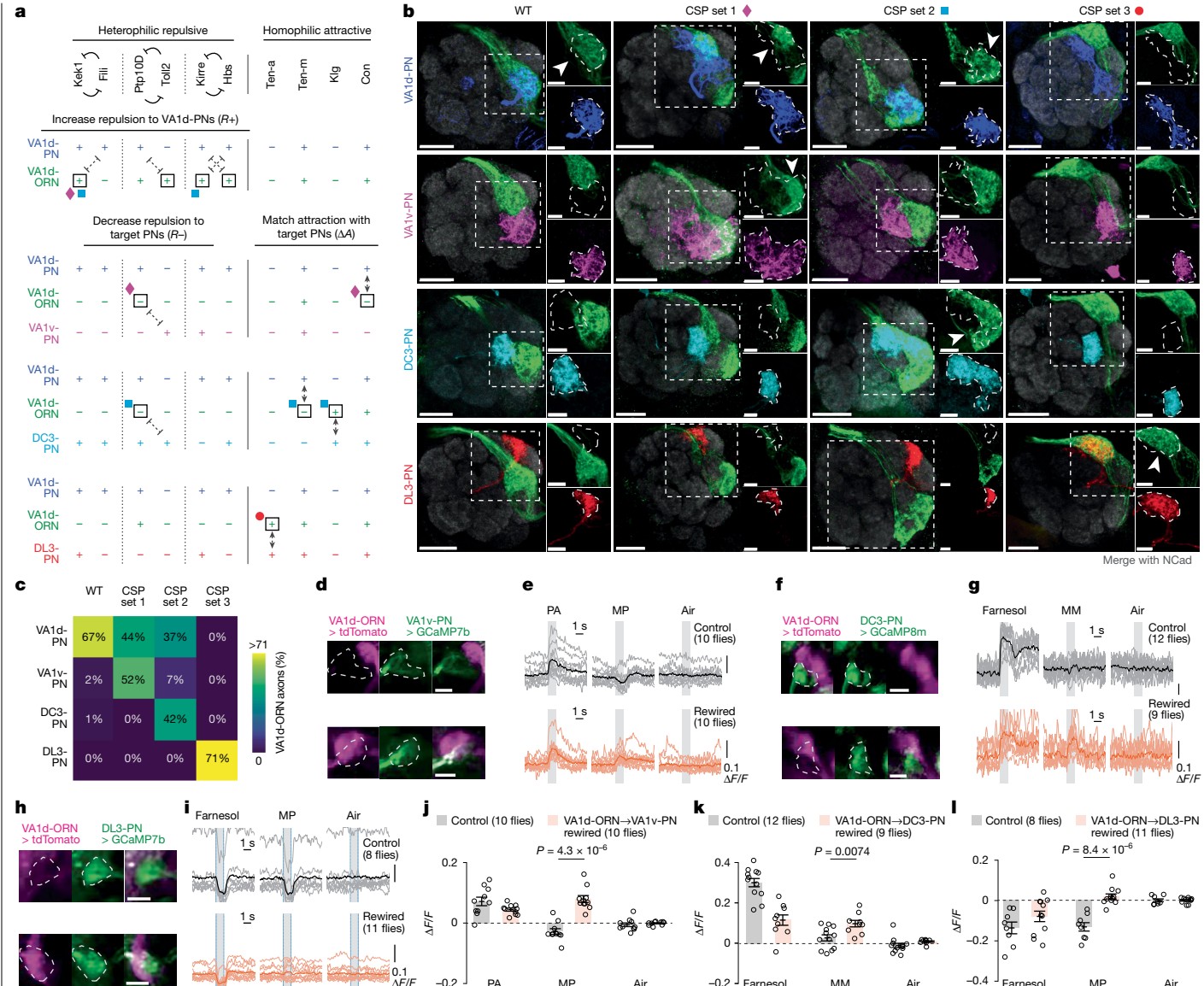

**Fig. 5 | Rewiring VA1d-ORNs with distinct PN partners following the same manipulation strategies. a**, Summary of expression levels of the ten CSPs at 24–30 h APF and the three manipulation strategies in the experiments of rewiring VA1d-ORNs (green) away from VA1d-PNs (blue) and towards VA1v-PNs (magenta), DC3-PNs (cyan) or DL3-PNs (red), respectively. The same colour code is used in **b**. Nomenclature as in Fig. 1d–g. Available protein expression data are in Extended Data Fig. 9. Diamonds, squares and circles indicate the genetic manipulations used in the final switches of VA1d-ORNs to VA1v-PNs, DC3-PNs and DL3-PNs, respectively. **b**, Maximum *z*-projections of adult antennal lobes around VA1d-ORN axons in the wild type and in three rewiring experiments. In each row, VA1d-ORNs are co-labelled with one PN type according to labels on the left. In each column, the same set of genetic manipulations is used. CSP set 1, Kek1 OE + *Ptp10D* RNAi + *Con* RNAi + *Sema2b* RNAi; CSP set 2, Kek1 OE + Kirre OE + *Ptp10D* RNAi + *Ten-m* RNAi + Klg OE; CSP set 3, Ten-a OE + *Sema2b* RNAi. *Sema2b* RNAi is used for rerouting the VA1d-ORN trajectory to better match the dendrites of VA1v-PNs and DL3-PNs. The borders of the dendrites of each PN type are shown with dashed lines. Arrowheads indicate the overlap of VA1d-ORN axons with dendrites of specific PN types. Insets are magnifications of the dashed squares to their left. Scale bars, 20 μm (main images); 10 μm (insets). **c**, Percentage of VA1d-ORN axons overlapping with the dendrites of each PN type (indicated on the left) in the wild type and in three rewiring conditions (indicated at the top). See Extended Data Fig. 10 for *n* values. **d**, Images of tdTomato signal in VA1d-ORN axons and GCaMP7b signal in VA1v-PN dendrites from a control

fly (top) and a VA1d-ORN→VA1v-PN fly (bottom). Images are averaged across the entire recording. The VA1v glomerulus is outlined according to the GCaMP7b signal. Imaging set-up as in Fig. 3. **e**, Averaged GCaMP7b activity in VA1v-PN dendrites in response to odorized air flows. Grey vertical stripes indicate odour deliveries (1 s each). Light and dark traces indicate the means of individual flies and the population mean, respectively. In controls, the fly pheromone PA specifically activates VA1v-ORNs[12] and the fly pheromone MP mainly activates VA1d-ORNs[28]. **f,g**, As in **d,e**, but for the rewiring of VA1d-ORNs to DC3-PNs instead of VA1v-PNs. GCaMP8m is used instead of GCaMP7b. In controls, the odorant farnesol mainly activates DC3-ORNs[36]. The fly pheromone MM specifically activates VA1d-ORNs and VA1v-ORNs[12] and is used here instead of MP because we observed a positive response of DC3-PNs to MP in the control. **h,i**, As in **d,e**, but for the rewiring of VA1d-ORNs to DL3-PNs instead of VA1v-PNs. Farnesol is used as a control because DL3-specific odorants remain unknown. In the control fly image, the VA1d-ORN signal is absent because VA1d-ORN axons and DL3-PN dendrites occupy different *z*-positions from the current imaging perspective. Scale bars, 20 μm (**d,f,h**). **j–l**, Change of GCaMP activity in dendrites of VA1v-PNs (GCaMP7b; **j**), DC3-PNs (GCaMP8m; **k**) and DL3-PNs (GCaMP7b; **l**) to odorized air flow. The change of activity is calculated by subtracting the average GCaMP activity in the 0.5 s before the odour onset from that in the last 0.5 s of odorized air flow. Circles indicate the means of individual flies; bars indicate the population mean ± s.e.m. Same fly numbers as in **e,g,i**, respectively. Two-sided unpaired *t*-test.

## Discussion

We have shown here that the fly olfactory circuit can be to a large extent rewired when two to five CSPs are changed in a single ORN type (Figs. 1, 2 and 5). This occurred even though dozens of CSPs are differentially expressed between different ORN types during the synaptic partner matching period (Extended Data Fig. 3). The rewiring expanded the physiological response to odours in downstream PNs (Figs. 3 and 5) and altered the courtship behaviour in one case (Fig. 4).

The CSP combinatorial code for rewiring should be closely related—if not identical—to the CSP code used during natural wiring. To illustrate this, consider the DA1-ORN→VA1v-PN rewiring. First, the five CSPs involved in the rewiring are differentially expressed between DA1-ORNs and VA1v-ORNs (Extended Data Fig. 3). The directions of gene-expression manipulation—whether up or down—match the discrepancy between these two ORN types. Second, both loss- and gain-of-function manipulations in most of the five CSPs alone significantly decreased the matching of DA1-ORN axons with DA1-PN dendrites or caused a mismatch of DA1-ORN axons with VA1v-PN dendrites (Extended Data Fig. 4), suggesting that these CSPs are involved in distinguishing the wiring specificity of DA1-ORNs and VA1v-ORNs naturally. Finally, rewiring leads to a gain of function at both the physiological and the behavioural level, pointing to its potential utility in an evolutionary context.

The fact that the rewiring was successful despite our lack of precise control over the level and timing of the CSP manipulations suggests that the combination of key CSPs is more crucial than the exact levels and timing of their expression. This is consistent with the general notion that many biological systems are robust in their tolerance to variations in gene expression. The precision of rewiring could be further improved if we can better control our genetic manipulations in level and timing, and by manipulating additional CSPs that we might have missed (for example, in the case of VA1d-ORN→DA1-PN and VA1d-ORN→DC3-PN rewiring in Fig. 5).

Our results show that synaptic partner matching seems to be flexible in the specific CSPs used, as long as they execute a common set of strategies: matching attractive CSPs between partners; avoiding repulsive CSPs between partners; and displaying repulsive CSPs between non-partners. Furthermore, CSPs of different families[5,6,17,18]—those containing immunoglobulin-like domains (Klg, Kirre, Hbs and Kek1), leucine-rich repeats (Con, Fili, Kek1 and Toll2), fibronectin III domains (Ptp10D and Hbs), and teneurin (Ten-a, Ten-m)—work together in different combinations for synaptic partner matching at different glomeruli. We speculate that these protein families converge onto common intracellular signalling pathways to regulate cytoskeletal changes that underlie attraction[4] and repulsion. We further note that protein motifs in these CSPs, and in many cases individual CSPs themselves, are evolutionarily conserved across invertebrates and vertebrates[2,8]. Thus, the combinatorial action of different CSP types we described here could be used to control synaptic partner matching in nervous systems from insects to mammals.

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

# Methods

## Fly husbandry and stocks

Flies were reared on a standard cornmeal medium at 25 °C under a 12-h–12-h light–dark cycle. To enhance transgene expression levels, flies from all genetic perturbation experiments, including control groups, were shifted to 29 °C shortly before puparium formation. Detailed genotypes for each experiment are listed in Supplementary Table 1.

## Molecular cloning and generation of transgenic flies

To generate QF2 lines, we used pENTR/D-TOPO vectors with various enhancer insertions (gifts from the laboratory of G. Rubin) as entry vectors for Gateway cloning into the pBPQF2Uw vector using LR Clonase II Enzyme mix (Invitrogen, 11791020). pBPQF2Uw was made using NEBuilder HiFi DNA assembly master mix (New England Biolabs) to replace the GAL4 on the pBPGAL4.2Uw-2 vector (Addgene, 26227) with QF2 from pBPGUw-HACK-QF2 (Addgene, 80276). The resulting constructs were sequence-verified and inserted into JK22C landing sites by Bestgene. pGP-5XQUAS-IVS-Syn21-jGCaMP8m-p10 was made using NEBuilder HiFi DNA assembly master mix (New England Biolabs) to replace the 20XUAS on the pGP-20XUAS-IVS-Syn21-jGCaMP8m-p10 vector (Addgene, 162387) with 5XQUAS from pQUAST (Addgene, 24349). Plasmids were injected to embryos at BestGene. Genetic labelling with these drivers is unlikely to disrupt normal development; a previous study showed that drivers with improved translation efficiency could increase GFP expression by 20-fold with no apparent effect on neuronal morphology[37].

## Immunostaining

The procedures used for fly dissection, brain fixation and immunostaining were described previously[10]. For primary antibodies, we used rat anti-DNcad (1:30, DSHB, RRID AB_528121), chicken anti-GFP (1:1,000, Aves Labs, RRID AB_10000240), rabbit anti-DsRed (1:500, Takara Bio, RRID AB_10013483) and mouse anti-rat CD2 (1:200, Bio-Rad, OX-34).

## Confocal imaging

Immunostained brains were imaged using a laser-scanning confocal microscope (Zeiss LSM 780). Images of antennal lobes were taken as confocal stacks with 1-mm-thick sections. Representative single sections were shown to illustrate the arborization features of ORN axons and PN dendrites, with brightness adjustment, contrast adjustment and image cropping done in ImageJ.

## Calculating the percentage of ORN axons matching with PN dendrites

PN dendritic pixels and ORN axonal pixels were defined by first smoothening the image using 'gaussian blur' (radius = 2 pixels) and then thresholding the image based on the algorithm 'Otsu' in Fiji. We found that this algorithm could efficiently separate the neurons of interest from the background. Irrelevant signals (such as the PN axons, cell bodies or autofluorescence) that still persisted after these operations were manually masked out in the analysis. A portion of ORN axons were considered as matching with PN dendrites if they had overlapping pixels on a single z-plane in the image. Note that the definition of glomerulus becomes vague as ORN axons and PN dendrites innervate more and more outside the original glomerulus.

The calculated overlap between ORN axons and PN dendrites is always lower than 100%. This is because ORN axons or PN dendrites do not occupy the entire glomerulus, for a technical reason and for a biological reason. Technically, if one examines axons and dendrites with super resolution, they should not overlap at all, because each physical space should be occupied by only one entity if the resolution is sufficiently high. In our quantifications, we used 'gaussian blur' to best recapitulate the adjacent areas of a single axon or dendrites that should be considered as 'overlap'. This is an empirical parameter and would

not achieve 100% overlap. Biologically, as well as ORN–PN synapses, both ORNs and PNs also form reciprocal synapses with antennal-lobe LNs. Regions with ORN–LN synapses lack PN dendrites; regions with PN–LN synapses lack ORN axons. Thus, ORN axons and PN dendrites don't overlap in these regions.

In our analyses, we use the same parameters to quantify all genetic conditions. Thus, our conclusions about the changing of ORN–PN overlap under different genetic conditions should not be affected by these factors.

## Ca²⁺ imaging and data analysis

**Delivery of odour stimuli.** Ten microlitres of PA (Thermo Fisher Scientific, 376910010) or ten microlitres of cVA (Cayman Chemical, 10010101) was applied to filter paper (Amazon, B07M6QJ2JX) inserted inside a 1-ml pipette tip. The pipette tip was left aside for at least 30 min before being positioned approximately 5 mm away from the fly antenna. Close positioning of PA and cVA is necessary because both odorants are large pheromone molecules with relatively low volatility. This method has also been used in other studies[12]. Other odorants, such as MP (Thermo Fisher Scientific, L05509.36), MM (Thermo Fisher Scientific, 165015000) and farnesol (Thermo Fisher Scientific, 119121000), were stored in a small glass bottle and delivered to the fly antenna through tubing, with a 10% dilution in heavy mineral oil on the day of experiments. A constant stream of charcoal-filtered air (1 l per min) was directed towards the fly, switching to odorant-containing air for 1 s as the odour stimulus before returning to the airstream. A pulse of charcoal-filtered air served as a negative control. Odorants, including the control pulse, were interleaved with at least 15-s intervals. Each odorant was delivered two to three times per recording, with the delivery sequence shuffled within each cycle. As described previously[38], we glued flies to a custom stage. Dissection and imaging protocols also followed a previous study[38].

**Data acquisition and alignment.** We used a two-photon microscope with a moveable objective (Ultima IV, Bruker). The two-photon laser (Chameleon Ultra II Ti:Sapphire, Coherent) was tuned to 925 nm in all of the imaging experiments. We used a ×16/0.8 NA objective (Nikon) for all imaging experiments. The laser intensity at the sample was 15–30 mW. A 575-nm dichroic split the emission light. A 490–560-nm bandpass filter (Chroma) was used for the green channel and a 590–650-nm bandpass filter (Chroma) was used for the red channel. We recorded all imaging data using a single z-plane, at a rate of 9–13 Hz. We perfused the brain with extracellular saline composed of 103 mM NaCl, 3 mM KCl, 5 mM N-Tris(hydroxymethyl) methyl-2-aminoethanesulfonic acid (TES), 10 mM trehalose, 10 mM glucose, 2 mM sucrose, 26 mM NaHCO₃, 1 mM NaH₂PO₄, 1.5 mM CaCl₂ and 4 mM MgCl₂. All data were digitized by a Digidata 1550b digitizer (Molecular Devices) at 10 kHz, except for the two-photon images, which were acquired using PrairieView (Bruker) at varying frequencies and saved as TIFF files for later analysis. We used the frame triggers associated with our imaging frames (from Prairie View), recorded on the Digidata 1550b, to carefully align odorant delivery with Ca²⁺ imaging measurements.

**Image registration.** The image stacks were motion-corrected using non-rigid motion correction (NoRMCorre[39]) and then manually validated to check for motion artefacts.

**Defining regions of interest.** To analyse Ca²⁺ imaging data, we defined regions of interest (ROIs) in Fiji and Python for GCaMP signals from PN dendrites in one hemisphere, or both hemispheres when the PN dendritic signals were available. We treated the entire PN dendrites from one hemisphere as one ROI.

**Calculating fluorescence intensities.** We used ROIs, defined above, as the unit for calculating fluorescent intensities (see above). For each ROI, we calculated the mean pixel value at each time point and then

used the method $\Delta F/F_0$ to calculate, where $F_0$ is the mean of the lowest 5% of raw fluorescence values in a given ROI over time and $\Delta F$ is $F - F_0$.

## Courtship assay

Flies were collected shortly after eclosion. Male flies were housed individually, whereas female flies (Canton-S) were housed in groups of approximately ten. All females used as courtship targets were three-to-five-day-old virgins. All males tested in the experiments had not mated. Males were four to seven days old in Fig. 4 and Extended Data Fig. 8a–d and two days old in Extended Data Fig. 8e–i to lower the courtship baseline in males. All male flies were either $w^+$, or $w^-$ but carried more than three mini-white markers from the transgenes they possessed. In single-pair courtship assays, two males (or one male and one female) were introduced into a custom-made courtship chamber with a diameter of 2 cm. In the courtship chain assay, five DA1-ORN→VA1v-PN males were introduced into a custom-made courtship chamber with a diameter of 5 cm. Courtship experiments were performed under low white light to reduce baseline courtship activity, because vision is well known to influence the vigour of fly courtship. Before being placed into the courtship chamber, flies were briefly grouped in a tube and anaesthetized on ice for less than 10 s. Once placed into the chamber, most flies were able to move immediately but did not fly away. Fly behaviour was recorded for more than 25 min with a video camera at 13 frames per second, and the first 25 min were quantified. In the single-pair male–male courtship assay, a control male and a rewired male were age-matched, and one of them was marked with an oil paint marker (Sharpie) on their thorax at least one day before the experiment. The paint was alternated between control and rewired males. LED lights (660 nm) were used to activate DA1-ORNs expressing csChrimson.

## Statistics and reproducibility

For the representative images from Fig. 1b,c and Extended Data Figs. 1a,b, 2c–f, 6c,d and 10a,b, at least five samples were examined with similar results.

## Fly study design

No statistical tests were used to determine sample size. We used sample sizes (around 6–20 flies per condition) that have been shown to have sufficient statistical power in similar experiments in the past. We did not exclude flies or data from any analysis, unless brains stained for imaging appeared unsuitable (for example, broken) at the time of imaging. All experiments discussed in the paper were performed on multiple flies, with the sample size specified. For most two-photon and behavioural experiments, data across multiple days were collected with consistent results. For immunostaining, data across multiple days were collected and all imaged brains showed the same qualitative pattern of staining. Organisms are not allocated to control and experimental groups by the experimenter in this work; rather, the genotypes of the flies determine their group. Thus, randomization of individuals into treatments groups is not relevant. The investigators were not blind to fly genotype. All data collection and analysis were done computationally. During this process, data from control groups and experimental groups were analysed equally using the same well-established protocols, reducing the influence of the investigator.

## Reporting summary

Further information on research design is available in the Nature Portfolio Reporting Summary linked to this article.

## Data availability

All data are included in the manuscript and its supplementary materials. Source data are provided with this paper.

## Code availability

Code is available on GitHub (https://github.com/Cheng-Lyu/CL_Stanford.git).

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

**Acknowledgements** We thank the Bloomington *Drosophila* Stock Center, the Vienna *Drosophila* Resource Center and the *Drosophila* Genetic Resource Consortium for fly stocks; G. Maimon for custom plates and T. Clandinin for extracellular saline used in Ca²⁺ imaging; H. Dionne and G. Rubin for enhancer plasmids; L. Jayne for assistance with some VA1d-ORN experiments; and T. Clandinin, K. Shen, G. Maimon, W. Qi and S. Sethi, as well as members of the L.L. laboratory, especially T. Hindmarsh Sten and D. Pederick, for discussions. This work was supported by the National Institutes of Health (R01-DC005982), Wu Tsai Neuroscience Institute of Stanford University (L.L.) and the Gatsby Foundation. C.L. was supported by the Stanford Science Fellows programme. L.L. is an investigator of Howard Hughes Medical Institute.

**Author contributions** C.L. and L.L. conceived the project. C.L. performed all of the experiments and analysed the data. C.L., Z.L. and L.L. jointly interpreted the data and decided on new experiments. Z.L. and C.X. assisted with cloning. J.K. assisted with behavioural experiments. C.L. and L.L. wrote the paper, with inputs from all of the other authors. L.L. supervised the work.

**Competing interests** The authors declare no competing interests.

**Additional information**
**Correspondence and requests for materials** should be addressed to Liqun Luo.

**a** Adult

Dorsal → Lateral

DA1-ORN axon

Antennal lobe

50 μm

R78H05-p65.AD, R22E04-GAL4.DBD > UAS-mCD8-GFP          Ncad

**b** 32 hours after puparium formation (h APF)

Dorsal → Lateral

20 μm

Antennal lobe

**Extended Data Fig. 1 | DA1-ORN split-GAL4 characterization. a**, In adults, the split-GAL4, R78H05-AD + R22E04-DBD, labels DA1-ORNs in the whole brain as revealed by GFP staining (green). The brain border is dash-outlined. **b**, At around 32 h APF, the same split-GAL4 driver most strongly labels DA1-ORNs (solid arrowhead) and weakly and sparsely labels a few other ORN types whose axons take the ventromedial trajectory (open arrowhead). Magenta: N-cadherin (Ncad) staining for neuropils. Maximum z-projections are shown.

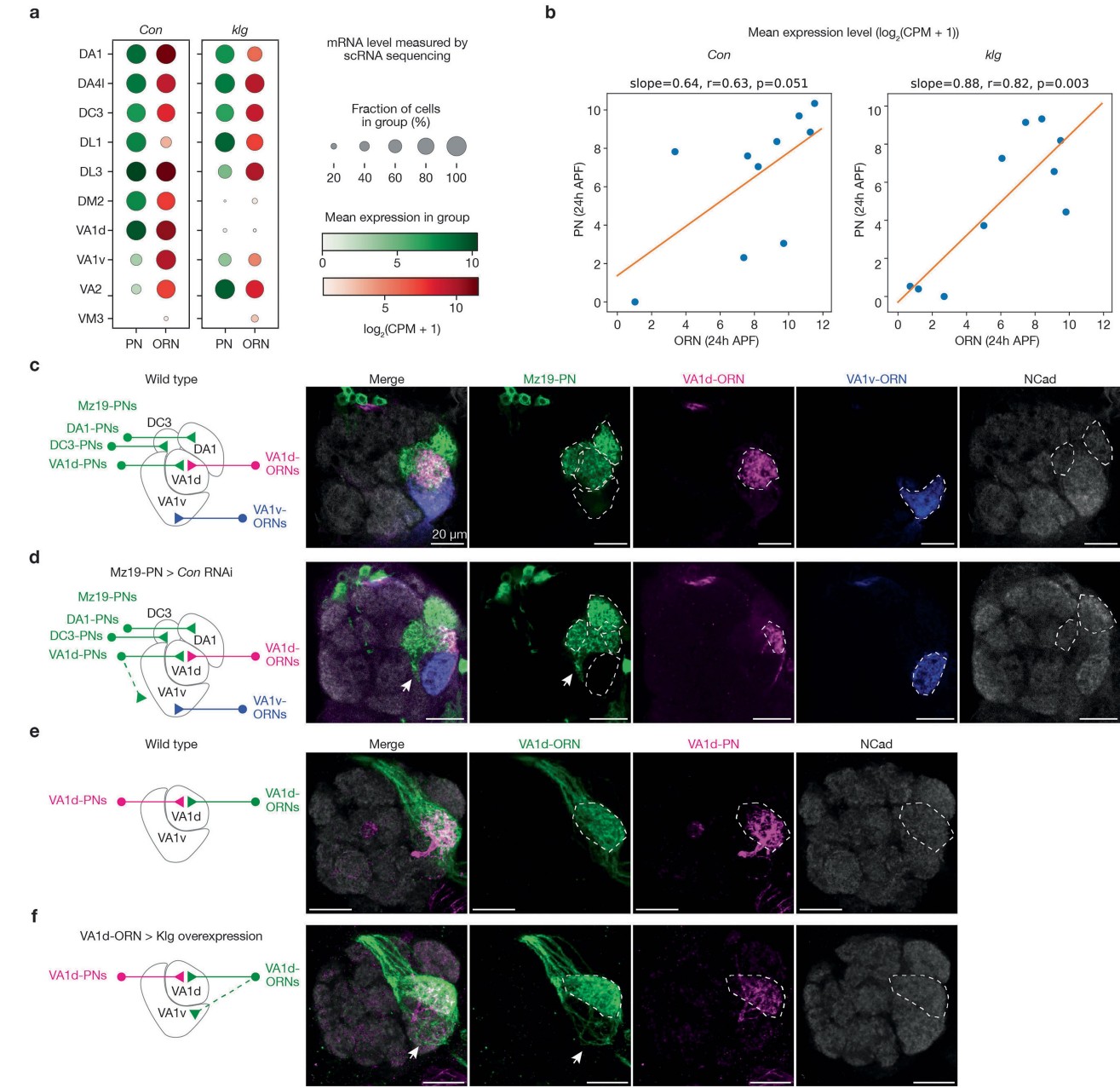

**Extended Data Fig. 2 | The CSPs Klg and Con regulate synaptic partner matching in the fly olfactory circuit.** Klingon (Klg) and connectin (Con) are homophilic adhesion molecules and have previously been reported to regulate the wiring of the *Drosophila* visual circuit and neuromuscular system, respectively[17,18]. Here, both the expression pattern and the genetic manipulation results suggest that Klg and Con regulate synaptic partner matching of the *Drosophila* olfactory circuits. **a,b**, At around 24 h APF, *Con* and *Klg* exhibit matching expression patterns across ORN and PN types based on scRNA-seq data[19,20]. Dot plot (**a**) and scatter plot (**b**) with linear fitting (orange solid line) are shown. Two-sided linear regression was used. No adjustment was made. Blue dots in **b** represent the glomerular types shown in **a**. **c**, Maximum projection of optical sections of the same antennal lobes from a wild-type brain. DA1-PNs, DC3-PNs, and VA1d-PNs (green) are labelled by GFP using *Mz19-GAL4*. VA1d-ORNs (magenta) are labelled by tdTomato using the *Or88a* promoter. VA1v-ORNs (blue) are labelled by rat CD2 using the *Or47b* promoter. The borders of DA1 and DC3 glomeruli are outlined based on the N-cadherin (NCad) staining

signal. The VA1v and VA1d glomeruli are outlined according to the ORN signals. The glomerular targeting of these neurons in the wild-type brain is summarized by the schematic on the left. Scale bar = 20 μm in this and all other panels. **d**, Same as **c**, but expressing *con* RNAi in the three Mz19+ PN types. The dendrites of some PNs, probably VA1d-PNs based on anatomical tracing, ectopically target outside glomeruli (arrows). 14 out 14 antennal lobes show similar phenotype using RNAi line 17898 from Vienna *Drosophila* Resource Center and 6 out of 14 antennal lobes show similar phenotype using RNAi line 28967 from Bloomington *Drosophila* Stock Center. **e**, Maximum projection of optical sections of the same antennal lobe from a wild-type brain, with VA1d-ORNs labelled using GFP (green) by GAL4/UAS and VA1d-PNs labelled using tdTomato (magenta) by QF2/QUAS. The border of VA1d glomerulus is outlined based on the N-cadherin (NCad) staining signal. The VA1d-ORN axons and VA1d-PN dendrites match well. **f**, Same as **e**, but overexpressing Klg in VA1d-ORNs. The axons of some VA1d-ORNs ectopically mistarget to the VA1v glomerulus (arrows).

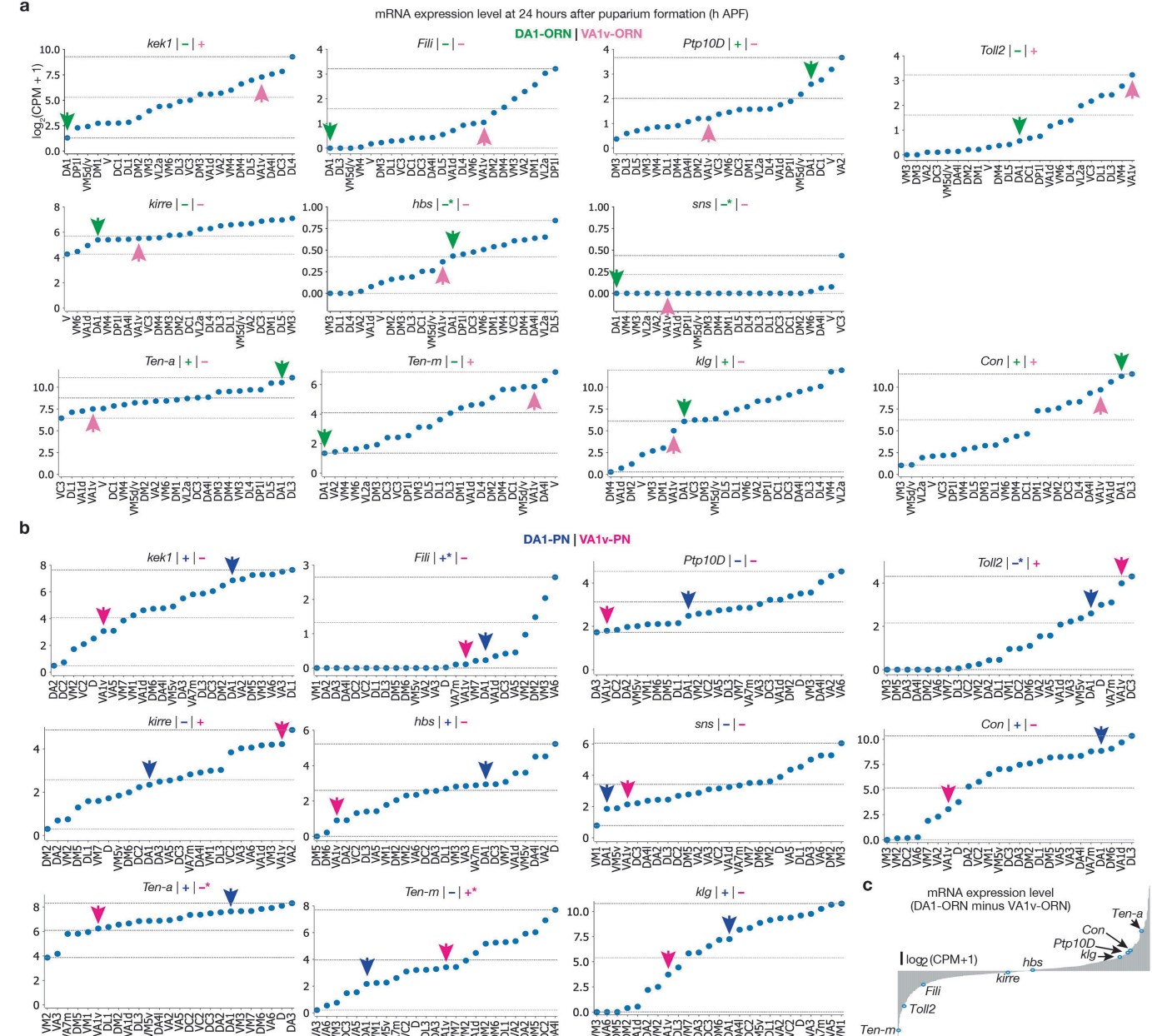

**Extended Data Fig. 3 | Expression levels of the CSPs in the developing**
**_Drosophila_ olfactory system, used in the DA1-ORN rewiring experiments.**
Here we provide the basis of assigning '+' or '−' for the expression levels of CSPs
in Fig. 1d. **a**, mRNA expression levels of the wiring molecules used in the DA1-ORN
rewiring experiments. The expression levels are based on the scRNA-seq data[19,20]
and all the ORN types decoded are shown. Plots are generated using data at ~24 h
APF in this and all other panels. In each subplot, dashed horizontal lines represent
the lowest, the highest, and the median (the average of the minimum and
maximum) expression levels. The green arrow indicates the data from DA1-ORNs,
and the magenta arrow indicates the data from VA1v-ORNs. The '+' or '−' sign
indicates the expression level as inferred from the scRNA-seq data based on
whether the expression level is above ('+') or below ('−') the median. Because
the scRNA-seq data are prone to measurement noise and may not accurately
reflect protein expression due to post-transcriptional regulation, we corrected
expression levels using the protein data and in vivo genetic manipulation
results in CSPs where additional data were available. *designates places where

corrections about the '+' or '−' are made, and the sign showed here is after the
correction. Hbs and Sns are considered lowly expressed in both ORN types
because of the absolute expression levels of these two mRNAs are very low.
The unit of the y axis is log₂(counts per million read + 1) in this and all other panels.
**b**, Same as **a**, but plotting the expression level in all the PN types decoded.
The blue arrow indicates the data from DA1-PNs and the magenta arrow indicates
the data from VA1v-PNs. Fili is considered highly expressed in DA1-PNs based on
the data from a previous study (in Fig. 3d)[9]. Toll2 is considered lowly expressed in
DA1-PNs based on the conditional-tag data from the companion study (Fig. 1d)[5].
Ten-a is considered lowly expressed in VA1v-PNs and Ten-m is considered highly
expressed in VA1v-PNs based on the antibody staining data from a previous study
(Fig. 2)[6]. **c**, The difference of mRNA expression level of CSPs that are expressed
between DA1-ORNs and VA1v-ORNs, with the 10 wiring molecules used in the
rewiring experiments indicated. *Sticks and stones* (*sns*), encoding a second
interacting partner of kirre in addition to Hbs (companion study[5]), is not
included here because it is lowly expressed in both ORN types.

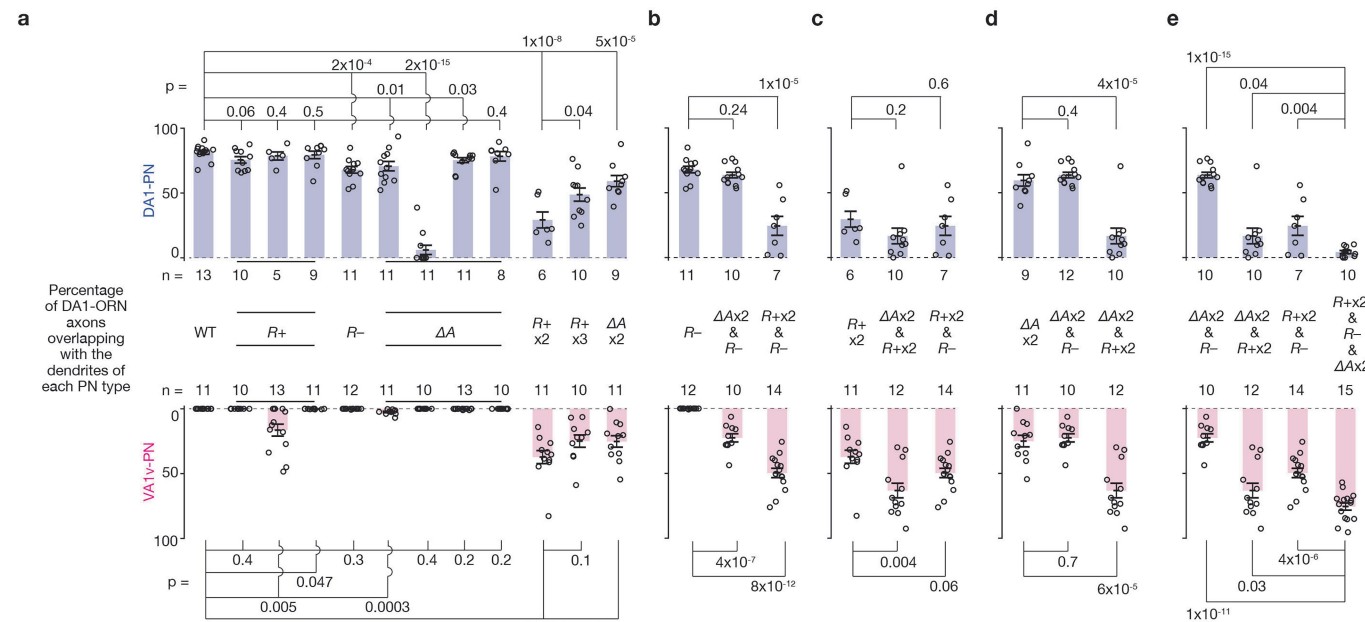

**Extended Data Fig. 4 | Statistical tests in the DA1-ORN→VA1v-PN rewiring.**
Same plots as in Fig. 2a, but with statistical tests added. Percentage of DA1-ORN axons overlapping with the dendrites of DA1-PNs (top) and VA1v-PNs (bottom). Circles indicate individual antennal lobes; bars indicate the population mean ± s.e.m. '*R+*': Kek1 overexpression (OE), Fili OE, and Kirre OE (left to right). '*R−*': *Ptp10D* RNAi. '*ΔA*': *ten-a* RNAi, Ten-m OE, *klg* RNAi, and *con* RNAi (left to right). '*R+* x2': Kek1 OE + Fili OE. '*R+* x3': Kek1 OE + Fili OE + Kirre OE. '*ΔA* x2': *ten-a* RNAi + *con* RNAi. **a**, All statistical tests were performed between 'WT' and an individual manipulation condition, respectively, except the one test where it is performed between '*R+* x2' and '*R+* x3'. In all the eight single-CSP manipulations, six—Fili OE, Kirre OE, *Ptp10D* RNAi, *ten-a* RNAi, Ten-m OE, and *klg* RNAi—showed observable yet very mild DA1-ORN→VA1v-PN mismatching phenotypes, with some reaching statistical significance whereas others not (n.s.). Two-sided

t-test was used for all tests except the one test performed between '*R+* x2' and '*R+* x3' where one-sided t-test was used. This is because the null hypothesis of this test is that the '*R+* x3' group does not cause stronger mismatching phenotype compared to the '*R+* x2' group, which is directional. **b**, All tests were performed between '*R−*' and other manipulation conditions, respectively. Two-sided t-test was used. **c**, All tests were performed between '*R+* x2' and other manipulation conditions, respectively. Two-sided t-test was used. **d**, All tests were performed between '*ΔA* x2' and other manipulation conditions, respectively. Two-sided t-test was used. **e**, All tests were performed between '*R+* x2 & *R−* & *ΔA* x2' (the DA1-ORN rewired condition) and other manipulation conditions, respectively. One-sided t-test was used because the null hypothesis is that the rightmost group does not show stronger mismatching phenotype compared to the groups to the left, which is a directional hypothesis.

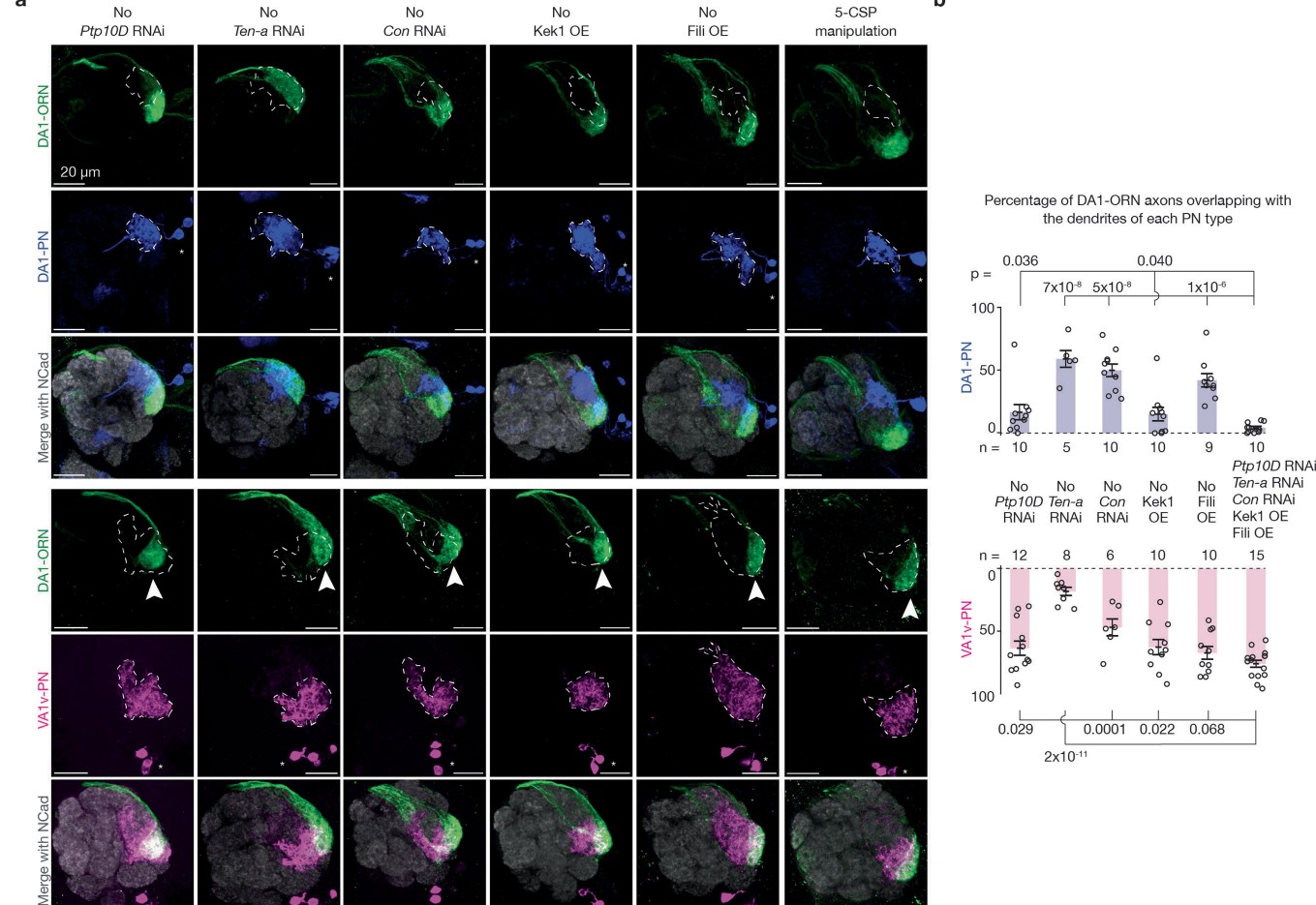

**Extended Data Fig. 5 | Omitting any one of the five CSP changes reduces the magnitude of DA1-ORN rewiring. a**, Genetic manipulations are labelled on the top. Maximum z-projections of adult antennal lobes around DA1-ORN axons (green) are shown. Top three rows: DA1-PNs (blue) are co-labelled with borders dashed outlined. Bottom three rows: VA1v-PNs (magenta) are co-labelled with borders dashed outlined. The genetic manipulation condition for the rightmost column contains five genetic changes as listed, same as the rewiring condition as in Fig. 2c. Genetic manipulation conditions for the left five columns are the same as for the rightmost column except missing one manipulation as indicated. Arrowheads indicate the mismatch of DA1-ORN axons with VA1v-PN dendrites; scale bar = 20 μm; * designates PN cell bodies. **b**, Percentage of DA1-ORN axons overlapping with the dendrites of DA1-PNs (top) and VA1v-PNs (bottom). Same genetic manipulation conditions as in **a**. Circles indicate individual antennal lobes; bars indicate the population mean ± s.e.m. All the tests are performed between the data from the rightmost column and the data from other columns, respectively. Two-sided unpaired t-test is used.

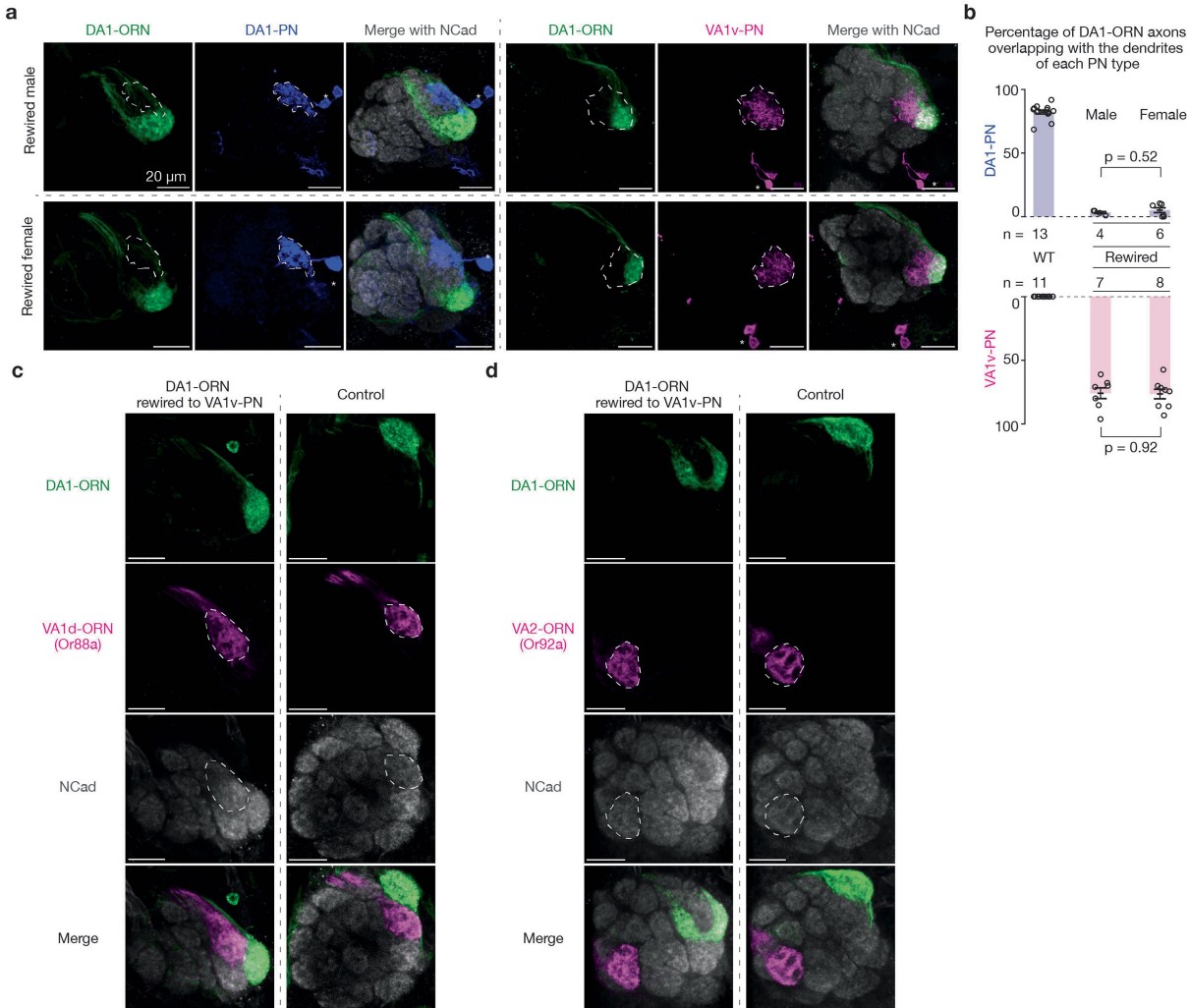

**Extended Data Fig. 6 | Males and females exhibit similar DA1-ORN→VA1v-PN rewiring and the axons of some non-DA1 and non-VA1v ORNs remain confined in their original glomeruli. a**, Maximum z-projections of adult antennal lobes around DA1-ORN axons (green) are shown. Top row: rewired male; bottom row: rewired female. Left three columns: DA1-PNs (blue) are co-labelled with borders dashed outlined; right three columns: VA1v-PNs (magenta) are co-labelled with borders dashed outlined. The genetic manipulation condition is the same as the rewiring condition as in Fig. 2c. Scale bar = 20 μm here and throughout the figure; *designates PN cell bodies. **b**, Percentage of DA1-ORN axons overlapping with the dendrites of DA1-PNs (top) and VA1v-PNs (bottom). Circles indicate individual antennal lobes; bars indicate the population mean ± s.e.m. Two-sided unpaired t-tests are performed. **c**, Here we tested whether in DA1-ORN→VA1v-PN rewired

flies, VA1d-ORN axons target normally to the VA1d glomerulus, which is in between DA1 and VA1v. Shown are maximum z-projection of adult antennal lobes around DA1-ORN (green, labelled by a membrane-targeted GFP driven by a split-GAL4) and VA1d-ORN (magenta, *Or88a* promotor driving tdTomato) axons. Grey: N-cadherin (Ncad) staining for neuropils. Scale bar = 20 μm. The VA1d glomerulus boarders are dash-outlined based on VA1d-ORN signals. The axons of VA1d-ORNs remain confined to their endogenous glomerulus. Note ventral shift of DA1-ORN axon terminals in rewired flies compared with control, consistent with their matching with VA1v-PN dendrites. **d**, Same as **c**, but with VA2-ORN co-labelled using Or92a promotor driving rat CD2, showing that they are confined within the VA2 glomerulus (outlined) in the rewired fly as in control.

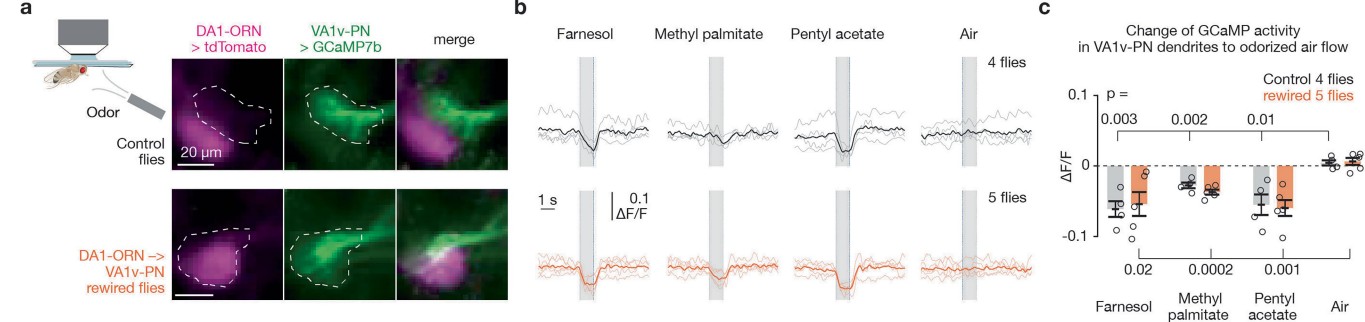

**Extended Data Fig. 7 | VA1v-PNs in DA1-ORN-rewired flies retain similar levels of inhibitory response to non-cognate odours tested. a**, Imaging neural activity in a plate-tethered fly with odorized air flow delivered to the fly antennae. Images of tdTomato signal in DA1-ORN axons and GCaMP7b signal in VA1v-PN dendrites are shown from a control fly (top) and a DA1-ORN rewired fly (bottom). Images are averaged across the entire recording. The VA1v glomerulus is outlined according to GCaMP7b signal. Scale bar = 20 µm. **b**, Averaged GCaMP7b activity in VA1v-PN dendrites in response to odorized air flows, measured by fluorescence intensity change over baseline (ΔF/F). Top: control flies. Bottom: DA1-ORN rewired flies. The grey vertical stripes indicate odorized air flows (1 s each). Light-coloured traces indicate the means of individual flies; dark-coloured traces indicate the population mean. Farnesol strongly activates DC3-ORNs[36] (spatially close to DA1 and VA1v glomeruli), fly

pheromone MP mainly activates VA1d-ORNs[28] (in between DA1 and VA1v glomeruli), and pentyl acetate activates a variety of types of ORNs[40]. **c**, Change of GCaMP7b activity in VA1v-PN dendrites to odorized air flows. The change of activity is calculated by subtracting the average GCaMP7b activity in the 0.5 s before the onset of odour delivery from that in the 0.4 s flanking the offset of odour delivery. Circles indicate the means of individual flies; bars indicate the population mean ± s.e.m. VA1v-PNs in DA1-ORN rewired flies retain similar levels of inhibitory response to non-cognate odours tested as in wild-type flies. Note that the inhibitory response to MP is weaker than the response to other two odours, consistent with the low volatility of the large molecular weight of MP. P < 0.02 for all odorized responses to the control air group. Two-sided unpaired t-tests were used.

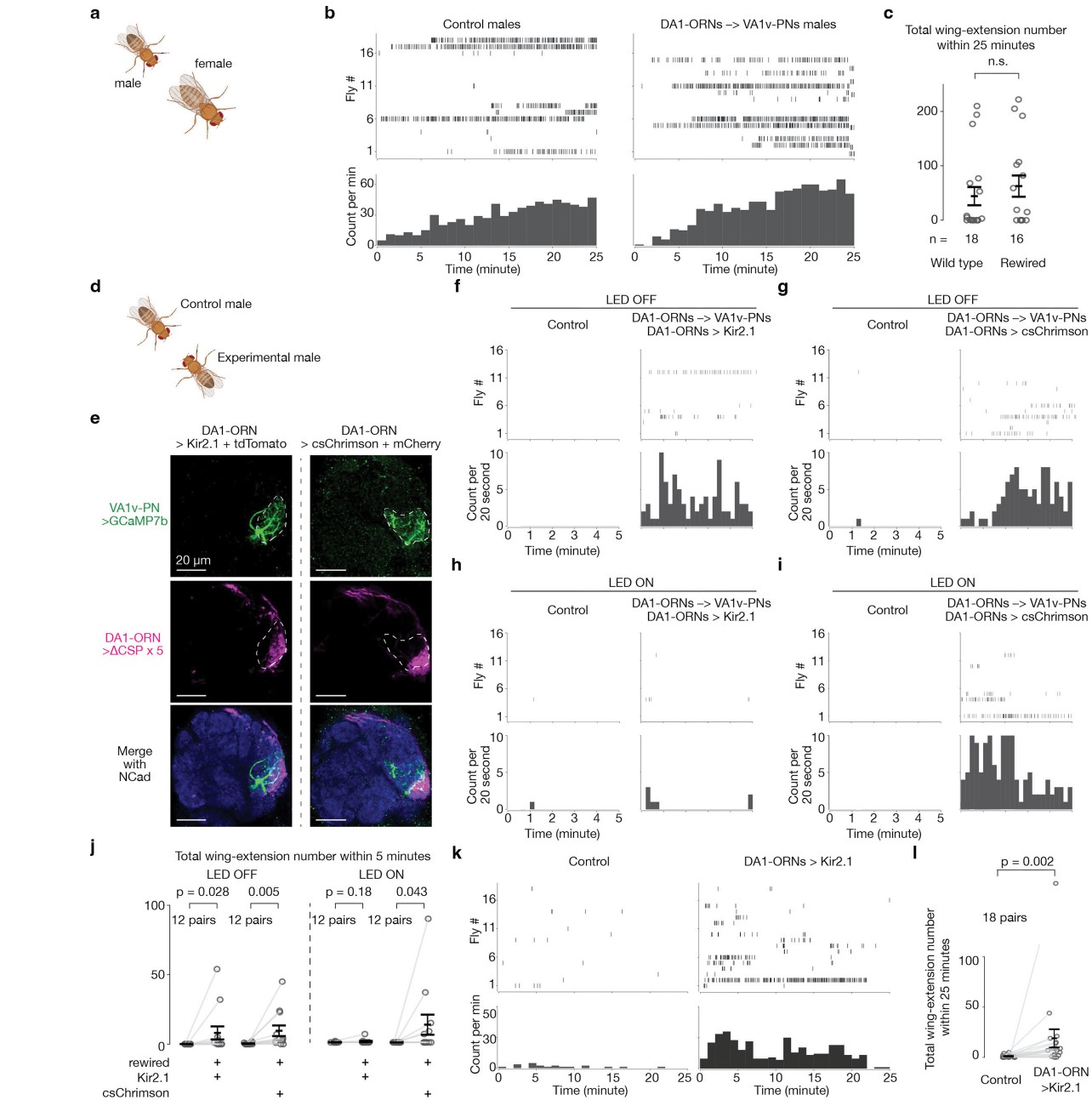

**Extended Data Fig. 8** | See next page for caption.

**Extended Data Fig. 8 | Further behavioural examination of the courtship activity of DA1-ORN→VA1v-PN-rewired males. a**–**c**, Wild-type and DA1-ORN→VA1v-PN rewired males exhibit similar courtship activity towards virgin females. **a**, Courtship assay where one male and one virgin female are introduced in the same behavioural chamber to monitor the courtship activity from the male towards the female. The chamber diameter is 2 cm. **b**, Unilateral wing extension rasters (top) and extension count per minute (1-min bins, bottom). Left: wild-type males; right: DA1-ORN rewired males. **c**, Total wing extension number during the 25-minute recordings. Circles indicate individual flies; bars indicate the population mean ± s.e.m. Two-sided Wilcoxon signed-rank test is used given the non-normal distribution of the data points. **d**–**l**, In DA1-ORN→VA1v-PN rewired males, two connectivity changes could potentially contribute to behavioural changes: (1) DA1-ORNs losing endogenous connection to DA1-PNs (loss of connection or LoC) and (2) DA1-ORNs gain new connection to VA1v-PNs (gain of connection or GoC). Two experiments here tested if both LoC and GoC contribute to the increased male–male courtship in rewired flies. **d**, Courtship assay where one wild-type male and one experimental male are introduced in the same behavioural chamber to monitor their courtship activity towards each other. Two-day old males are used here and throughout the figure to lower the courtship baseline in males. **e**, Maximum z-projection of adult antennal lobes around DA1-ORN axons (magenta, labelled by a membrane-targeted tdTomato driven by a split-GAL4). DA1-PN dendrites are co-labelled (green, labelled by GCaMP driven by a LexA driver) with borders dash-outlined. Scale bar = 20 μm. DA1-ORNs also express the same 5 cell-surface-proteins (ΔCSP x 5) as the rewired flies in Fig. 2d. The rewiring of DA1-ORN axons to VA1v-PN dendrites still persist with exogenously expressing csChrimson (a red-shifted channelrhodopsin for activating neurons when LED is on) or Kir2.1 (an inward rectifying K⁺ channel for silencing neuronal activity) across development. **f**, Same as **b**, but for the male–male courtship assay where one male is the control and the other male has DA1-ORNs rewired to VA1v-PNs and silenced by additional expression of Kir2.1. The recording duration was reduced to 5 minutes. Red LEDs are turned off. **g**, Same as **f**, but the experimental male is changed to one with DA1-ORNs rewired to VA1v-PNs and exogenously expressing csChrimson. **h**,**i**, Same as **f**,**g**, but with LEDs turned on to activate the males expressing csChrimson. **j**, Total wing extension number during the 5-minute recordings. Circles indicate individual flies; bars indicate the population mean ± s.e.m. Two-sided Wilcoxon signed-rank test is used given the non-normal distribution of the data points. **k**,**l**, Same as **b**,**c**, but for the male–male courtship assay where one male is the control and the other male with DA1-ORNs silenced by exogenously expressing Kir2.1. (1) To test the contribution LoC, we compared the courtship activity of wild-type males versus DA1-ORN rewired but silenced (by exogenously expressing inward-rectifier potassium channel Kir2.1) males. If LoC contributed to increased courtship activity in rewired males, we should expect to see DA1-ORN rewired but silenced males show stronger courtship activity than wild-type males. Results **f**,**j** support this working model. This result is not surprising, because silencing DA1-ORNs alone yields a similar behavioural phenotype (**k**,**l** and ref. 13 using *Or67d* mutant). (2) To test the contribution of GoC, we note that DA1-ORN rewired flies simultaneously have LoC. Therefore, we sought to compare the difference between flies with 'GoC & LoC' (males with DA1-ORN rewired and can be optogenetically activated) and flies with 'LoC' alone (males with DA1-ORN rewired but silenced). Stronger courtship activity in flies with 'GoC & LoC' would supports that 'GoC' has a positive contribution on rewired male's enhanced courtship activity. Results **f**–**j** support this working model. When LEDs were turned off, rewired males from both groups showed stronger courtship activity over wild-type males, supporting that 'LoC' alone could increase male courtship activity towards other males. When red LEDs were turned on, the courtship difference in 'LoC' group disappeared. Because neither male from this group exogenously expressed any light-sensitive channels, we hypothesized that bright lighting might inhibit male courtship activity. However, in males with DA1-ORN rewired and exogenously expressing csChrimson, they exhibited significantly stronger courtship activity over wild-type males despite the lighting effect. This supports that activation of DA1-ORNs in rewired males contribute to the increase of male–male courtship activity.

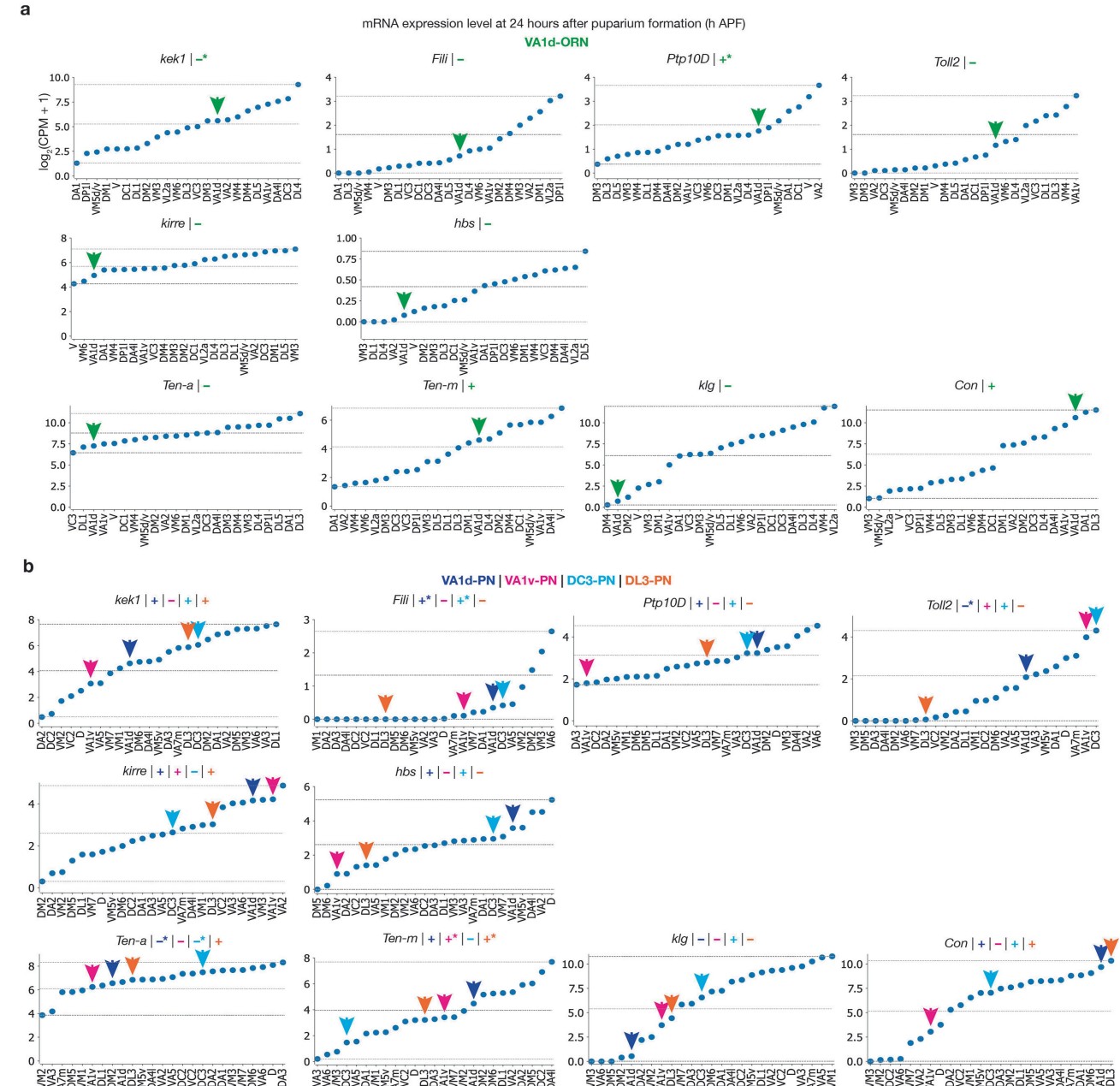

**Extended Data Fig. 9 | Expression levels of the ten CSPs in the VA1d-ORN rewiring.** Here we provide the basis of assigning '+' or '−' for the expression levels of CSPs in Fig. 5a. **a**, mRNA expression levels of the 10 CSPs used in the VA1d-ORN rewiring experiments. The expression levels are based on the scRNA-seq data[19,20] and all the ORN types decoded are shown. Plots are generated using data at 24 h APF) in this and all other panels. In each subplot, the lowest and highest expression levels are indicated by the dashed horizontal lines. The green arrow indicates the data from VA1d-ORNs. The '+' or '−' sign indicates the expression level as inferred from the scRNA-seq data based on whether the expression level is above ('+') or below ('−') the median, which is the average of the minimum and maximum value. Because the scRNA-seq data are prone to measurement noise and may not accurately reflect protein expression due to post-transcriptional regulation, we corrected the RNA data using the protein data and the in vivo genetic manipulation results in CSPs where additional data were available. *designates places where corrections are made, and the sign shown here is after

the correction. Kek1 is considered lowly expressed based on the conditional-tag data from the companion study (Extended Data Fig. 2)[5]. Fili is considered highly expressed in DA1-PNs based on the data from a previous study (in Fig. 3d)[9]. Because the RNA level in DC3-PNs is higher than in DA1-PNs, we considered Fili also highly expressed in DC3-PNs. Ptp10D is considered highly expressed based on the conditional-tag data as well as the knockout experiments from the companion study (Figs. 1c and 2c)[5]. The unit of the y axis is log₂(counts per million read + 1). **b**, Same as **a**, but plotting the expression level in all the PN types decoded. The blue arrow indicates the data from DA1-PNs, the magenta arrow indicates the data from VA1v-PNs, light blue arrow indicates the data from DC3-PNs, and the orange arrow indicates the data from DL3-PNs. Toll2 is considered lowly expressed in VA1d-PNs based on the conditional-tag data from the companion study (Fig. 1d)[5]. Ten-a is considered lowly expressed in VA1d- and DL3-PNs, whereas Ten-m is considered highly expressed in VA1v- and DL3-PNs based on the antibody staining data from a previous study (Fig. 2)[6].

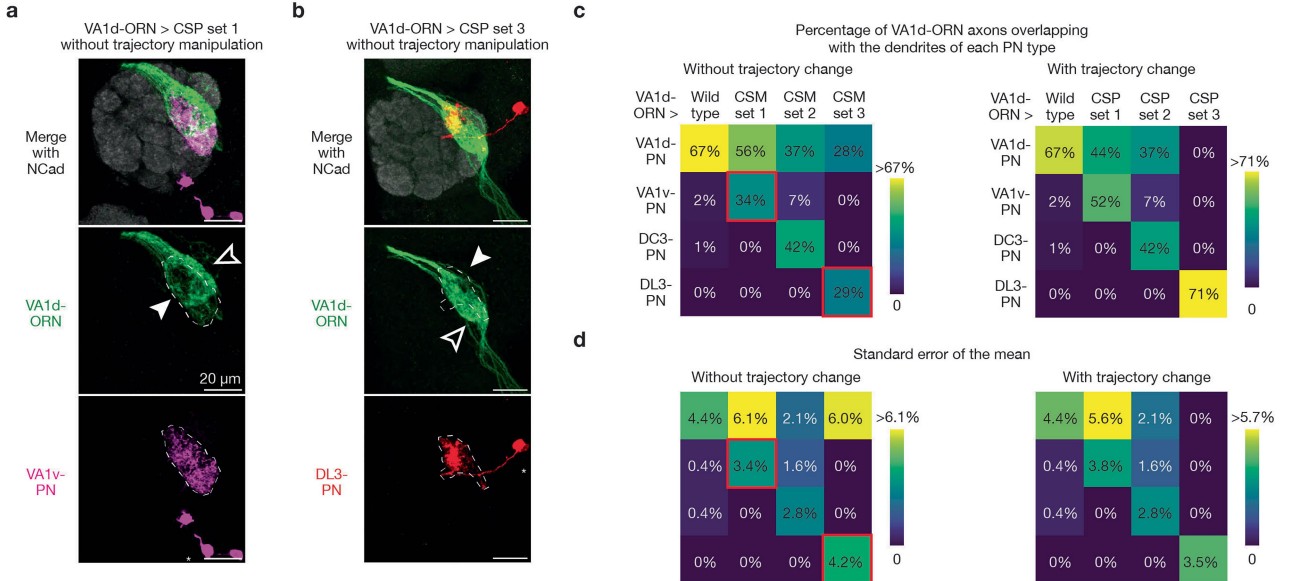

**a**

VA1d-ORN > CSP set 1
without trajectory manipulation

Merge with NCad

VA1d-ORN

20 μm

VA1v-PN

**b**

VA1d-ORN > CSP set 3
without trajectory manipulation

Merge with NCad

VA1d-ORN

DL3-PN

**c**

Percentage of VA1d-ORN axons overlapping
with the dendrites of each PN type

Without trajectory change

| VA1d-ORN > | Wild type | CSM set 1 | CSM set 2 | CSM set 3 |
|---|---|---|---|---|
| VA1d-PN | 67% | 56% | 37% | 28% |
| VA1v-PN | 2% | 34% | 7% | 0% |
| DC3-PN | 1% | 0% | 42% | 0% |
| DL3-PN | 0% | 0% | 0% | 29% |

>67% ... 0

With trajectory change

| VA1d-ORN > | Wild type | CSP set 1 | CSP set 2 | CSP set 3 |
|---|---|---|---|---|
| VA1d-PN | 67% | 44% | 37% | 0% |
| VA1v-PN | 2% | 52% | 7% | 0% |
| DC3-PN | 1% | 0% | 42% | 0% |
| DL3-PN | 0% | 0% | 0% | 71% |

>71% ... 0

**d**

Standard error of the mean

Without trajectory change

| 4.4% | 6.1% | 2.1% | 6.0% |
|---|---|---|---|
| 0.4% | 3.4% | 1.6% | 0% |
| 0.4% | 0% | 2.8% | 0% |
| 0% | 0% | 0% | 4.2% |

>6.1% ... 0

With trajectory change

| 4.4% | 5.6% | 2.1% | 0% |
|---|---|---|---|
| 0.4% | 3.8% | 1.6% | 0% |
| 0.4% | 0% | 2.8% | 0% |
| 0% | 0% | 0% | 3.5% |

>5.7% ... 0

**Extended Data Fig. 10 | Trajectory manipulation of VA1d-ORN axons improves rewiring with the dendrites of VA1v- and DL3-PNs. a**, Maximum z-projections of adult antennal lobes around VA1d-ORN axons in the rewiring experiments without trajectory manipulation of VA1d-ORN axons. Genetic manipulations = Kek1 overexpression (OE) + *Ptp10D* RNAi + *con* RNAi. VA1v-PNs (magenta) are co-labelled with borders dash-outlined. The solid arrowhead indicates the mismatch of VA1d-ORN axons with VA1v-PN dendrites; the open arrowhead indicates the part of VA1d-ORN axons that do not match with VA1v-PN dendrites. Scale bar = 20 μm; *designates PN cell bodies. **b**, Same as **a**, but for the rewiring experiment of VA1d-ORN axons to DL3-PN dendrites. Genetic manipulations = Ten-a OE. DL3-PNs (red) are co-labelled with borders dash-outlined. The solid arrowhead indicates the mismatch of VA1d-ORN axons with DL3-PN dendrites; the open arrowhead indicates the part of VA1d-ORN axons that do not match with DL3-PN dendrites. **c**, Percentage of VA1d-ORN axons overlapping with the dendrites of each PN type (indicated on the left) in wild type and the three rewired conditions without (left) and with (right) trajectory manipulations. The right matrix is a repeat of Fig. 5c for ease of comparison. The two red boxes in the left matrix indicate the genetic manipulation conditions shown in **a** and **b**, respectively. Note that in these two boxes, the ratios of VA1d-ORN axons rewired to the target PNs are less compared to the conditions with trajectory manipulations, suggesting that trajectory manipulation of VA1d-ORN axons improves rewiring to the dendrites of VA1v- and DL3-PNs. The group with trajectory change, from left to right: VA1d-PN row, n = 9, 7, 14, 20; VA1v-PN row, n = 6, 12, 12, 10; DC3-PN row, n = 6, 10, 20, 12; DL3-PN row, n = 9, 12, 10, 11. The group without trajectory change, from left to right: VA1d-PN row, n = 9, 7, 14, 20; VA1v-PN row, n = 6, 17, 12, 10; DC3-PN row, n = 6, 10, 20, 12; DL3-PN row, n = 9, 12, 10, 12. **d**, Same as **c**, but plotting the s.e.m. instead of the population mean.

# Reporting Summary

## Statistics

For all statistical analyses, confirm that the following items are present in the figure legend, table legend, main text, or Methods section.

| n/a | Confirmed | |
|---|---|---|
| ☐ | ☒ | The exact sample size (*n*) for each experimental group/condition, given as a discrete number and unit of measurement |
| ☐ | ☒ | A statement on whether measurements were taken from distinct samples or whether the same sample was measured repeatedly |
| ☐ | ☒ | The statistical test(s) used AND whether they are one- or two-sided<br>*Only common tests should be described solely by name; describe more complex techniques in the Methods section.* |
| ☐ | ☒ | A description of all covariates tested |
| ☐ | ☒ | A description of any assumptions or corrections, such as tests of normality and adjustment for multiple comparisons |
| ☐ | ☒ | A full description of the statistical parameters including central tendency (e.g. means) or other basic estimates (e.g. regression coefficient) AND variation (e.g. standard deviation) or associated estimates of uncertainty (e.g. confidence intervals) |
| ☐ | ☒ | For null hypothesis testing, the test statistic (e.g. *F*, *t*, *r*) with confidence intervals, effect sizes, degrees of freedom and *P* value noted<br>*Give P values as exact values whenever suitable.* |
| ☒ | ☐ | For Bayesian analysis, information on the choice of priors and Markov chain Monte Carlo settings |
| ☒ | ☐ | For hierarchical and complex designs, identification of the appropriate level for tests and full reporting of outcomes |
| ☒ | ☐ | Estimates of effect sizes (e.g. Cohen's *d*, Pearson's *r*), indicating how they were calculated |

*Our web collection on statistics for biologists contains articles on many of the points above.*

## Software and code

Policy information about availability of computer code

| Data collection | Two-photon imaging data were collected using PrairieView 5.4 (Bruker). Frame triggers and olfactory stimulus data were recorded as voltages on a Digidata 1550b (Molecular Devices) I/O board. Immunostained brains were imaged using a laser-scanning confocal microscope (Zeiss LSM 780). |
|---|---|
| Data analysis | Two photon imaging data were motion-corrected using non-rigid motion correction (NoRMCorre, v0.1.1) and then pre-processed using Fiji (to define regions of interest, version: 2.1.0/1.54j). All data were analyzed with custom code in python 2.7 and 3.6 and are available on github (https://github.com/Cheng-Lyu/CL_Stanford.git). |

For manuscripts utilizing custom algorithms or software that are central to the research but not yet described in published literature, software must be made available to editors and reviewers. We strongly encourage code deposition in a community repository (e.g. GitHub). See the Nature Portfolio guidelines for submitting code & software for further information.

## Data

Policy information about availability of data

All manuscripts must include a data availability statement. This statement should provide the following information, where applicable:
- Accession codes, unique identifiers, or web links for publicly available datasets
- A description of any restrictions on data availability
- For clinical datasets or third party data, please ensure that the statement adheres to our policy

> All data are included in the manuscript and supplementary materials.

## Research involving human participants, their data, or biological material

Policy information about studies with human participants or human data. See also policy information about sex, gender (identity/presentation), and sexual orientation and race, ethnicity and racism.

| | |
|---|---|
| Reporting on sex and gender | Not applicable. |
| Reporting on race, ethnicity, or other socially relevant groupings | Not applicable. |
| Population characteristics | Not applicable. |
| Recruitment | Not applicable. |
| Ethics oversight | Not applicable. |

Note that full information on the approval of the study protocol must also be provided in the manuscript.

# Field-specific reporting

Please select the one below that is the best fit for your research. If you are not sure, read the appropriate sections before making your selection.

☒ Life sciences ☐ Behavioural & social sciences ☐ Ecological, evolutionary & environmental sciences

For a reference copy of the document with all sections, see nature.com/documents/nr-reporting-summary-flat.pdf

# Life sciences study design

All studies must disclose on these points even when the disclosure is negative.

| | |
|---|---|
| Sample size | No statistical tests were used to determine sample size. We used sample sizes (~6-20 flies per condition) that been previously shown to have sufficient statistical power in similar experiments in the past (e.g., Hong, Mosca, Luo 2012, Lyu, Abbott, Maimon 2022) |
| Data exclusions | We did not exclude flies or data from any analysis, unless brains stained for imaging appeared unsuitable (e.g., broken) at the time of imaging. |
| Replication | All experiments discussed in the paper were conducted on multiple animals with sample size specified. For most two-photon and behavioral experiments, data across multiple days were collected and the data across days were consistent. In immunohistochemistry plots, data across multiple days were collected and all imaged brains showed the same qualitative pattern of staining. |
| Randomization | Organisms are not allocated to control and experimental groups by the experimenter in this work, rather the flies' genotype determines their group. Thus, randomization of individuals into treatments groups is not relevant. |
| Blinding | The investigators were not blind to the flies' genotypes. All data collection and analysis were done computationally. During this process, data from control groups and experimental groups were analyzed equally using the same well-established protocols, therefore are less prone to investigator influence. |

# Reporting for specific materials, systems and methods

We require information from authors about some types of materials, experimental systems and methods used in many studies. Here, indicate whether each material, system or method listed is relevant to your study. If you are not sure if a list item applies to your research, read the appropriate section before selecting a response.

## Materials & experimental systems

| n/a | Involved in the study |
|-----|----------------------|
| ☐ | ☒ Antibodies |
| ☒ | ☐ Eukaryotic cell lines |
| ☒ | ☐ Palaeontology and archaeology |
| ☐ | ☒ Animals and other organisms |
| ☒ | ☐ Clinical data |
| ☒ | ☐ Dual use research of concern |
| ☒ | ☐ Plants |

## Methods

| n/a | Involved in the study |
|-----|----------------------|
| ☒ | ☐ ChIP-seq |
| ☒ | ☐ Flow cytometry |
| ☒ | ☐ MRI-based neuroimaging |

## Antibodies

| | |
|---|---|
| Antibodies used | rat anti-DNcad (from DSHB, RRID # AB_528121), chicken anti-GFP (from Aves Labs, RRID # AB_10000240), rabbit anti-DsRed (from Takara Bio, RRID # AB_10013483), and mouse anti-rat CD2 (1:200; OX-34, Bio-Rad) |
| Validation | All antibodies used in this study were validated as described at the following websites (and references therein): DSHB: https://dshb.biology.uiowa.edu/DN-Ex-8, Rockland: https://rockland-inc.com/store/Antibodies-to-GFP-and-Antibodies-to-RFP-600-901-215-O4L_23908.aspx, Takara: https://www.takarabio.com/products/antibodies-and-elisa/fluorescent-protein-antibodies/red-fluorescent-protein-antibodies?srsltid=AfmBOopUZqVextBqypoqsvRxsHH-H9rGlg0NFICn1UMie592NHF348BQ, Bio-Rad: https://www.bio-rad-antibodies.com/monoclonal/rat-cd2-antibody-ox-34-mca154.html?f=purified |

## Animals and other research organisms

Policy information about studies involving animals; ARRIVE guidelines recommended for reporting animal research, and Sex and Gender in Research

| | |
|---|---|
| Laboratory animals | We used male and female Drosophila melanogaster. All fly genotypes are described in details in the Methods. For behavioral experiments, female flies of the Canton-S strain were used, aged 3–5 days. In all other experiments, the w[1118] strain was used, with ages ranging from the pupal stage to 7 days old. |
| Wild animals | The study did not involve wild animals. |
| Reporting on sex | Most experiments were performed on both sexes and reached similar conclusion. Only the male-specific courtship behavioral experiments were performed using males. |
| Field-collected samples | The study did not involve samples collected from the field. |
| Ethics oversight | No ethical oversight was required because no vertebrates were used. |

Note that full information on the approval of the study protocol must also be provided in the manuscript.

## Plants

| | |
|---|---|
| Seed stocks | Not applicable. |
| Novel plant genotypes | Not applicable. |
| Authentication | Not applicable. |

