## [Peer Review File · Nature]

Rewiring an olfactory circuit by altering combinatorial cell-surface code

Corresponding Author: Dr Liqun Luo

Version 0:

Reviewer comments:

Referee #1

(Remarks to the Author)

This is a remarkable paper, in that the authors have figured out how to rewire a neural circuit using the information they have obtained over the years about the cell recognition molecules involved in antennal lobe wiring. No-one has done this before, to my knowledge.

I only have a few minor comments.

There are many typos and misspellings in the figures and text. They also need to provide more information about the odorants that are used in Figure 5.

Referee #2

(Remarks to the Author)

The study by Dr. Luo and colleagues addresses the question of how neural circuits are precisely assembled during development. Based on the functional roles of cell-surface proteins in synaptic partner matching presented in the companion manuscript, they systematically altered the combination of differentially expressed CSPs in DA1 ORNs and found a strategy to successfully rerouted their axons from the DA1 to the VA1v glomerulus. This success indicates that the field is closer to a comprehensive understanding of synaptic partner matching in neural circuit assembly, and this is likely to be a milestone in the field of neural development. Although I am enthusiastic about the study, I have major concerns about the conclusion that “the rewiring expanded the physiological response to odors in downstream PNs and altered male flies’ courtship behavior”. This needs further experiments to provide a tight link to the rewired ORNs.

1. While anatomical analysis supports the conclusion that DA1 ORNs have been rerouted to the VA1v glomerulus, it is unclear whether this glomerulus also receives other ORN types besides the endogenous VA1v ORNs and the rewired DA1 ORNs. Further investigation is needed to establish a causal relationship between the altered courtship behavior and the rewired DA1 ORNs. This can be determined by using a large repertoire of odor stimulus in the calcium imaging experiments presented in Figure 3. Specifically, if the responses of VA1v PNs in the rewired flies are limited to the ligands of the Or67d and Or47b receptors, then the rewiring is limited to just DA1 ORNs. Otherwise, the rewiring strategy has side effects on the targeting of other ORN types, which will need more care for data interpretation and require other behavioral tests to determine the relevance of the rewired DA1 ORNs (see below).

2. It is unclear whether the enhanced male-male courtship of the rewired males is related to cVA. To address this question, the authors need to show this does not happen to males that do not carry cVA. The data presented in Figure 4 comparing wildtype males with the rewired males does not rule out the possibility that other odors carried by the wildtype males, such as food odors, excite the VA1 PNs due to the rewiring of other ORN types to this glomerulus. Furthermore, the authors can investigate the courtship behavior of the rewired males toward virgin females and virgin females painted with cVA. These types of experiments will minimize the contribution of other ORNs that may have been rerouted to the VA1v glomerulus.

Referee #3

(Remarks to the Author)

Rewiring an olfactory circuit by altering the combinatorial code of cell-surface proteins

General comments

In this study Lyu and co-authors set out to identify the multiple molecular mechanisms by which synaptic specificity is achieved during development using as a model the antennal lobe of the fruitfly. One of the main difficulties in this type of study, as highlighted by the authors, is the high degree of redundancy of the wiring process: removing a single player usually has a limited impact on the outcome. To overcome this challenge they took a reductionist approach. By leveraging previous knowledge about the identity of several cell surface proteins (CSPs) involved in wiring, they were able to define a minimal set of genetic perturbations sufficient to rewire DA1 olfactory receptor neurons (ORNs) from their cognate projection neurons (PNs) to those of the VA1v glomerulus.

To test their hypothesis the authors generated complex animals allowing simultaneous 4 fold perturbations of DA1 ORNs interactions: introduce repulsion with DA1 PNs, remove repulsion with VA1v PNs, remove attraction to DA1 PNs and introduce attraction to VA1v PNs. These genetic tour de force and rigorous quantification convincingly demonstrated anatomical rewiring. The authors then used calcium imaging to prove that the rewiring led to functional connections and behavioural experiments to demonstrate that the new connections induced the expected behavioural switch: male flies started to court other males due to the re-routing of the male pheromone cVA, a male-male anti-aphrodisiac, into the VA1v, a sexually monomorphic fly odour channel.

Having achieved DA1 → VA1v rewiring the authors extended their results to a second type of ORN, VA1d, by rewiring it from VA1d PNs to DL3, DC3 and VA1v PNs. These results are compelling and, pleasingly, they required the introduction of an extra perturbation (Sema-2b), identified in the authors' previous work, to modify the axonal pathfinding process and allow the VA1d axons to encounter DL3 and VA1v dendrites. Here, for the first time, pathfinding and wiring specificity are both being manipulated precisely to achieve biologically relevant rewiring, as demonstrated by imaging calcium responses.

The insights gained in this study offer a framework to examine wiring specificity mechanisms across the brain, particularly by integrating the transcriptomics (revealing CSPs) and connectomics (offering endpoint wiring) atlases.

Specific comments

Text

Across the eight single-CSP manipulations (Fig. 1e–g), six showed observable but subtle DA1-ORNs→VA1v-PNs mismatching phenotypes (Fig. 1h, quantified in Fig. 2a and Extended Data Fig. 4)

This sentence is quite difficult to unpack from the complex figure cited, can the authors add the identity of the manipulations? Which are the six?

Notably, the axons of DA1-ORNs and VA1v-ORNs are segregated in the rewired flies (Fig. 2c), suggesting potential axon-axon repulsive interactions as previously shown in a different context.

This is an interesting and striking observation, did the authors find candidate CSPs that might mediate this segregation? If so, did they try to manipulate them?

This is consistent with our working model since a virgin female does not emit cVA, Could the authors change emit for have?

In all three rewiring experiments, the dendrites of target PNs gained response to VA1d-ORN-specific odors compared to in wild-type flies (Fig. 5d–l).

I agree with the statement for VA1v and DC3 but in the case of DL3, which is the one achieving the highest rewiring efficiency, there is no positive calcium signal (beyond the removal of the inhibition). Can the authors speculate why this is the case?

Can the authors mention in the methods if when measuring overlap they only consider the signal inside glomeruli or the entire neuronal arbour? Put differently, can cognate PN-ORN pairs reach c.100% overlap? (as would happen when considering only the glomerulus volume) or is it always lower because processes outside glomeruli don't overlap?

I found the discussion could do more to place the results within the broader context of known wiring mechanisms, particularly in vertebrates. Given Nature's broad readership, this would be especially valuable. The authors could also take the opportunity to speculate on how the mechanisms employed by different CSPs might interact—or remain functionally independent.

Figures

Figure 1

While the figures are overall very clear I find panels d–g quite complex, particularly leaving the + and - in e-g as in d and adding arrows. Could the authors swap the + and - as per the perturbation introduced and make them bold and bigger while removing the arrows? They could even put them in a different colour? Other ideas they might come up to make this easier to parse for the reader would be welcome.

Figure 2

It would be very helpful to have the gene perturbed indicated in each column of 2.a to avoid having to keep going back to the legend

Same for 2b

Figure 5

Red and magenta asterisk in 5.a are very hard to tell apart, can they be changed?

Extended data fig 3

Can the authors add a line to each plot indicating the midrange defining high and low expression? Also please add what

expression levels (+/-) you would assign to VA1v ORNs. This would be useful as a destination for the changes needed in DA1 ORNs for the rewiring. Could the authors also specify the origin of the protein data used to correct the scRNAseq predictions?

Version 1:

Reviewer comments:

Referee #1

(Remarks to the Author)

This is a remarkable paper. The other reviewers had some issues with the physiology and behavioral phenotypes, and I'd leave it up to them to decide if these have been addressed. I didn't have any criticisms of the original manuscript. This should definitely be published.

If there is space, the value of the paper to a wider audience would be increased by a cartoon figure illustrating all the changes. Fig. 5 has a lot of panels and takes a long time to work through.

Referee #2

(Remarks to the Author)

The authors have adequately addressed my concerns. I support the publication of the study.

Referee #3

(Remarks to the Author)

We appreciate the compliments from all three reviewers on the general interest and technical quality of our study, as well as their constructive criticisms. Below, we provide a point-by-point response to all comments by reviewers, with their reviews copied in blue.

Referee #1 (Remarks to the Author):

This is a remarkable paper, in that the authors have figured out how to rewire a neural circuit using the information they have obtained over the years about the cell recognition molecules involved in antennal lobe wiring. No-one has done this before, to my knowledge.

I only have a few minor comments.

There are many typos and misspellings in the figures and text. They also need to provide more information about the odorants that are used in Figure 5.

We thank the reviewer for appreciating the impact and novelty of our study. We have systematically streamlined our text and corrected typos that we can find throughout the paper. We have also provided more information about the odorants that are used in Figure 5 in the main text and the legend of Figure 5.

Referee #2 (Remarks to the Author):

The study by Dr. Luo and colleagues addresses the question of how neural circuits are precisely assembled during development. Based on the functional roles of cell-surface proteins in synaptic partner matching presented in the companion manuscript, they systematically altered the combination of differentially expressed CSPs in DA1 ORNs and found a strategy to successfully rerouted their axons from the DA1 to the VA1v glomerulus. This success indicates that the field is closer to a comprehensive understanding of synaptic partner matching in neural circuit assembly, and this is likely to be a milestone in the field of neural development. Although I am enthusiastic about the study, I have major concerns about the conclusion that “the rewiring expanded the physiological response to odors in downstream PNs and altered male flies’ courtship behavior”. This needs further experiments to provide a tight link to the rewired ORNs.

1. While anatomical analysis supports the conclusion that DA1 ORNs have been rerouted to the VA1v glomerulus, it is unclear whether this glomerulus also receives other ORN types besides the endogenous VA1v ORNs and the rewired DA1 ORNs. Further investigation is needed to establish a causal relationship between the altered courtship behavior and the rewired DA1 ORNs. This can be determined by using a large repertoire of

odor stimulus in the calcium imaging experiments presented in Figure 3. Specifically, if the responses of VA1v PNs in the rewired flies are limited to the ligands of the Or67d and Or47b receptors, then the rewiring is limited to just DA1 ORNs. Otherwise, the rewiring strategy has side effects on the targeting of other ORN types, which will need more care for data interpretation and require other behavioral tests to determine the relevance of the rewired DA1 ORNs (see below).

We appreciate the reviewer's enthusiasm about our study. Regarding the reviewer's concern: 'it is unclear whether this glomerulus also receives other ORN types besides the endogenous VA1v ORNs and the rewired DA1 ORNs', we think this is highly unlikely given that all of our genetic manipulation experiments (Figs. 1–4) were restricted to DA1-ORNs using a genetic driver that is highly specific to DA1-ORNs across development (Extended Data Fig. 1). We have emphasized this in multiple places in our revised manuscript to ensure that the reviewer (and our future readers) will not miss this important point.

Nevertheless, we appreciated the reviewer's rigor in further testing this and sought to examine potential changes in the connection from other ORNs (non-Or67d and non-Or47b) onto VA1v-PNs in the rewired males using both anatomical and physiological measures.

Anatomically, we labeled two non-Or67d, non-Or47b ORNs in both the DA1-ORN-rewired flies and control flies (new Extended Data Fig. 6). The axons of all the additional ORNs we examined remain confined to their endogenous glomeruli. This includes the VA1d glomerulus (targeted by ORNs expressing the odorant receptor Or88a) that is in between the DA1 and VA1v glomeruli.

Physiologically, as suggested by the reviewer, we examined whether DA1-ORN rewiring led to any changes in the response of VA1v-PNs to other odor stimulus using Ca^{2+} imaging experiments (new Extended Data Fig. 7). We tested three additional odors: farnesol that strongly activates DC3-ORNs (Ronderos et al., 2014, spatially close to DA1 and VA1v glomeruli), fly pheromone methyl palmitate (MP) that mainly activates VA1d-ORNs (Dweck et al., 2015, in between to DA1 and VA1v glomeruli), and pentyl acetate that activates a variety of ORN types (Hallem et al., 2006). Between control and DA1-ORN-rewired flies, we observed similar levels of inhibitory response in VA1v-PNs to the three odors tested (new Extended Data Fig. 7), consistent with the previously described lateral inhibition from local interneurons in the fly olfactory circuit (Olsen & Wilson 2008, Root et al., 2008). Note that the inhibitory response to MP is weaker than the response to other two odors, consistent with the low volatility of the large molecular weight of MP.

In summary, all our data support the working model that rewiring in this experiment is limited to DA1-ORNs and VA1v-PNs.

2. It is unclear whether the enhanced male-male courtship of the rewired males is related to cVA. To address this question, the authors need to show this does not happen to males that do not carry cVA. The data presented in Figure 4 comparing wildtype males with the rewired males does not rule out the possibility that other odors carried by the wildtype males, such as food odors, excite the VA1 PNs due to the rewiring of other ORN types to this glomerulus. Furthermore, the authors can investigate the courtship behavior of the rewired males toward virgin females and virgin females painted with cVA. These types of experiments will minimize the contribution of other ORNs that may have been rerouted to the VA1v glomerulus.

All of our new data described above support that DA1-ORN→VA1v-PN rewiring is limited to just DA1-ORNs but unlikely other ORN types. Nevertheless, we sought to further examine whether the enhanced male-male courtship in rewired males is related to the change of connection associated with DA1-ORNs and VA1v-PNs. Two connectivity changes could potentially contribute to behavioral changes: (1) DA1-ORNs losing endogenous connection to DA1-PNs (loss-of-connection or LoC) and (2) DA1-ORNs gain new connection to VA1v-PNs (gain-of-connection or GoC). To test if both LoC and GoC contribute to the increased male-male courtship in rewired flies (hence testing if the phenotype is related to cVA sensed by DA1-ORNs), we performed two sets of behavioral experiment described below (new Extended Data Fig. 8).

First, to test the contribution of LoC, we compared the courtship activity of wild-type males versus males with DA1-ORN rewired but silenced (by exogenously expressing inward-rectifier potassium channel Kir2.1). If LoC contributes to the increased courtship activity in rewired males, we should expect to see that DA1-ORN rewired but silenced males show stronger male-male courtship activity than control males. This was indeed the case (new Extended Data Fig. 8d–j). This result is not surprising, since silencing DA1-ORNs alone yields similar behavioral phenotype (Kurtovic et al., 2007 using Or67d mutant, and our own data described in the new Extended Data Fig. 8k–l using Kir2.1).

Second, to test the contribution of GoC, we note that DA1-ORN rewired flies would always have LoC. Therefore, we sought to compare the difference between flies with ‘GoC & LoC’ and flies with ‘LoC’ alone. If we see stronger courtship activity in flies with ‘GoC & LoC’, then it supports that ‘GoC’ also has a positive contribution on rewired male’s enhanced courtship activity. We performed experiments in two-day old males to lower the

courtship baseline in males (Lin et al., 2016). The ‘LoC’ alone group is males with DA1-ORN rewired but silenced versus control males. The ‘LoC & GoC’ group is males with DA1-ORN rewired and exogenously expressing csChrimson (that can be further optogenetically activated) versus control males. In the dark, rewired males from both groups show stronger courtship activity over wild-type males, consistent with that ‘LoC’ alone could increase male-male courtship activity. When red LEDs were turned on, the courtship difference in ‘LoC’ group disappeared (new Extended Data Fig. 8d–j). Since neither male from this group exogenously expresses any light-sensitive channels, we speculate that bright lighting may suppress male courtship activity. However, males with DA1-ORN rewired and expressing csChrimson still exhibit significantly stronger courtship activity towards wild-type males despite this lighting effect. This strongly supports that GoC also contributes to the increase of male-male courtship in DA1-ORN→VA1v-PN rewired flies. We believe this is equivalent to reviewer's suggestion of testing courtship with a male that cannot produce cVA, as here we are separately silencing and activating the cVA-sensing, DA1-ORNs.

Referee #3 (Remarks to the Author):

Rewiring an olfactory circuit by altering the combinatorial code of cell-surface proteins

General comments

In this study Lyu and co-authors set out to identify the multiple molecular mechanisms by which synaptic specificity is achieved during development using as a model the antennal lobe of the fruitfly. One of the main difficulties in this type of study, as highlighted by the authors, is the high degree of redundancy of the wiring process: removing a single player usually has a limited impact on the outcome. To overcome this challenge they took a reductionist approach. By leveraging previous knowledge about the identity of several cell surface proteins (CSPs) involved in wiring, they were able to define a minimal set of genetic perturbations sufficient to rewire DA1 olfactory receptor neurons (ORNs) from their cognate projection neurons (PNs) to those of the VA1v glomerulus.

To test their hypothesis the authors generated complex animals allowing simultaneous 4 fold perturbations of DA1 ORNs interactions: introduce repulsion with DA1 PNs, remove repulsion with VA1v PNs, remove attraction to DA1 PNs and introduce attraction to VA1v PNs. These genetic tour de force and rigorous quantification convincingly demonstrated anatomical rewiring. The authors then used calcium imaging to prove that the rewiring led to functional connections and behavioural experiments to demonstrate that the new connections induced the expected behavioural switch: male flies started to court other

males due to the re-routing of the male pheromone cVA, a male-male anti-aphrodisiac, into the VA1v, a sexually monomorphic fly odour channel.

Having achieved DA1 → VA1v rewiring the authors extended their results to a second type of ORN, VA1d, by rewiring it from VA1d PNs to DL3, DC3 and VA1v PNs. These results are compelling and, pleasingly, they required the introduction of an extra perturbation (Sema-2b), identified in the authors' previous work, to modify the axonal pathfinding process and allow the VA1d axons to encounter DL3 and VA1v dendrites. Here, for the first time, pathfinding and wiring specificity are both being manipulated precisely to achieve biologically relevant rewiring, as demonstrated by imaging calcium responses. The insights gained in this study offer a framework to examine wiring specificity mechanisms across the brain, particularly by integrating the transcriptomics (revealing CSPs) and connectomics (offering endpoint wiring) atlases.

We thank the reviewer for the appreciation of our work.

Specific comments

Text

Across the eight single-CSP manipulations (Fig. 1e–g), six showed observable but subtle DA1-ORNs→VA1v-PNs mismatching phenotypes (Fig. 1h, quantified in Fig. 2a and Extended Data Fig. 4)

This sentence is quite difficult to unpack from the complex figure cited, can the authors add the identity of the manipulations? Which are the six?

We have added the manipulation identities.

Notably, the axons of DA1-ORNs and VA1v-ORNs are segregated in the rewired flies (Fig. 2c), suggesting potential axon-axon repulsive interactions as previously shown in a different context.

This is an interesting and striking observation, did the authors find candidate CSPs that might mediate this segregation? If so, did they try to manipulate them?

We thank the reviewer for appreciating this axon-axon segregation phenomenon. We also share the interest of figuring out the underlying molecular mechanisms, but have not found the CSPs that might be responsible for this. We hope that we can address this question in future studies.

This is consistent with our working model since a virgin female does not emit cVA,
Could the authors change emit for have?

Done.

In all three rewiring experiments, the dendrites of target PNs gained response to VA1d-ORN-specific odors compared to in wild-type flies (Fig. 5d–l).

I agree with the statement for VA1v and DC3 but in the case of DL3, which is the one achieving the highest rewiring efficiency, there is no positive calcium signal (beyond the removal of the inhibition). Can the authors speculate why this is the case?

We speculate that the little positive response of DL3 PNs in rewired flies is because that in VA1d-ORN→DL3-PNs rewired flies, there still exists large amount of lateral inhibition onto DL3-PNs. One possibility is that MP might activate some other types of ORNs that in turn cancels out (via lateral inhibition) the positive synaptic transmission from VA1d-ORNs to DL3-PNs. We have added this speculation in the main text.

Can the authors mention in the methods if when measuring overlap they only consider the signal inside glomeruli or the entire neuronal arbour? Put differently, can cognate PN-ORN pairs reach c.100% overlap? (as would happen when considering only the glomerulus volume) or is it always lower because processes outside glomeruli don't overlap?

We have further described how we use quantify overlap and why it never reaches 100% in the Methods. In short, we quantify the overlap of ORN axons and PN dendrites (removing PN axons, cell bodies, etc). The quantification is not restricted to glomeruli since the definition of glomerulus becomes vague as ORN axons and PN dendrites innervate more and more outside the original glomerulus.

The calculated overlap between ORN axons and PN dendrites is always lower than 100% (see the data from our wild-type control). This is because ORN axons or PN dendrites do not occupy the entire glomerulus because of a technical reason and a biological reason. Technically, if one examines axons and dendrites with super resolution, they should not overlap at all as each physical space should be occupied by only one entity if the resolution is sufficiently high. In our quantifications, we used 'gaussian blur' to best recapitulate the adjacent areas of a single axon or dendrites that should be considered as 'overlap'. This is an empirical parameter and would not achieve 100% overlap. Biologically, besides ORN-PN synapses, both ORNs and PNs also form reciprocal synapses with antennal lobe local interneurons (LNs). Regions with ORN-LN synapses lack PN dendrites; regions with PN-LN

synapses lack ORN axons. Thus, ORN axons and PN dendrites don't overlap in these regions. We have added the above information in the relevant Methods section.

I found the discussion could do more to place the results within the broader context of known wiring mechanisms, particularly in vertebrates. Given Nature's broad readership, this would be especially valuable. The authors could also take the opportunity to speculate on how the mechanisms employed by different CSPs might interact—or remain functionally independent.

We thank the reviewer for these questions and suggestions. We have restructured the entire final paragraph of discussion so that our conclusions can be appreciated by Nature's broad readership. These include: (1) the general strategy attraction and repulsion for synaptic partner matching inferred from our rewiring experiments; (2) that CSPs of different families can be combinatorially used in for synaptic partner matching in different glomeruli; (3) that CSPs of different families likely converge onto common cytoskeletal signaling pathways underlying attraction because of (2); (4) that many of the CSP motifs and many individual CSPs themselves are conserved, suggesting that the combinatorial strategies we described could be widely used for synaptic partner matching from insects to mammals.

Figures

Figure 1

While the figures are overall very clear I find panels d–g quite complex, particularly leaving the + and - in e-g as in d and adding arrows. Could the authors swap the + and - as per the perturbation introduced and make them bold and bigger while removing the arrows? They could even put them in a different colour? Other ideas they might come up to make this easier to parse for the reader would be welcome.

We thank the reviewer for the suggestion. We have now deleted the arrows used to indicate manipulations, and replaced them with the '+' and '-' to indicate overexpression and knockdown perturbation conditions, respectively. We added a small grey box around each '+' or '-' to indicate experimental manipulations. We also applied the same scheme to Fig. 5a.

Figure 2

It would be very helpful to have the gene perturbed indicated in each column of 2.a to avoid

having to keep going back to the legend
Same for 2b

We have now added the gene perturbation into Fig. 2a&b.

Figure 5

Red and magenta asterisk in 5.a are very hard to tell apart, can they be changed?

We have now changed symbols for each PN type and adjusted the color scheme to better separate the two.

Extended data fig 3

Can the authors add a line to each plot indicating the midrange defining high and low expression? Also please add what expression levels (+/-) you would assign to VA1v ORNs. This would be useful as a destination for the changes needed in DA1 ORNs for the rewiring. Could the authors also specify the origin of the protein data used to correct the scRNAseq predictions?

We have now added a line to each plot indicating the median defining high and low expression, assigned expression level to VA1v-ORNs, and added further specifications on the origin of the protein data used to correct the scRNAseq predictions. We also applied the same edits to Extended Data Fig. 9.